# Migrating mesoderm cells self-organize into a dynamic meshwork structure during chick gastrulation

**Yukiko Nakaya, Mitsusuke Tarama, Sohei Tasaki, Ayako Isomura-Matoba, Tatsuo Shibata***

Laboratory for Physical Biology, RIKEN Center for Biosystems Dynamics Research, Kobe, Japan

**Abstract** Migration of cell populations is a fundamental process in morphogenesis and disease. The mechanisms of collective cell migration of epithelial cell populations have been well studied. It remains unclear, however, how the highly motile mesenchymal cells, which migrate extensively throughout the embryo, are connected with each other and coordinated as a collective. During chick gastrulation, cells emerging from the primitive streak and migrating in the 3D space between ectoderm and endoderm (mesoderm region) exhibit a novel form of collective migration. Using live imaging and quantitative analysis, such as topological data analysis (TDA), we found that these cells undergo a novel form of collective migration, in which they form a meshwork structure while moving away from the primitive streak. Overexpressing a mutant form of N-cadherin was associated with reduced speed of tissue progression and directionality of the collective cell movement, whereas the speed of individual cells remains unchanged. To investigate how this meshwork arises, we utilized an agent-based theoretical model, which suggests that cell elongation, cell-cell adhesion, and cell density are the key parameters for the meshwork formation. These data provide novel insights into how a supracellular structure of migrating mesenchymal cells may arise in loosely connected cell populations.

**\*For correspondence:**
tatsuo.shibata@riken.jp

**Competing interest:** The authors declare that no competing interests exist.

## Editor's evaluation

This important study describes a new mode of collective cell migration during chick embryo development. Quantitative live imaging revealed compelling evidence that cells self-organized into a 3D dynamic meshwork structure while migrating from the epiblast to the endoderm during gastrulation and that this network is associated with N-cadherin-mediated cell-cell adhesion. Agent-based simulations propose that cell-cell adhesions are required for the formation of the meshwork structure and that the cell aspect ratio and cell density may also play a role in the meshwork formation. This manuscript would be of interest to developmental and cell biologists as well as theoreticians studying tissue patterning and collective cell migration.

## Introduction

Collective behaviors of migrating cells are fundamental in processes of morphogenesis of tissues and organs, wound healing, and tumor metastasis. Such collective cell migration is typically found in epithelial tissues, in which the motile ability can be acquired by bringing multiple cells together into one group via intercellular adhesion (*Chuai and Weijer, 2009*; *Friedl and Gilmour, 2009*; *Friedl and Mayor, 2017*; *Sato et al., 2015*; *Scarpa and Mayor, 2016*). On the other hand, mesenchymal cells do not exhibit stable intercellular adhesion with surrounding cells, and they can migrate as individual

cells. However, even for these mesenchymal cells, collective cell migration that exploits transient cell-cell adhesion is required for the morphogenesis in living organisms (*Scarpa and Mayor, 2016*; *Shellard and Mayor, 2020*; *Theveneau and Mayor, 2014*). Neural crest (NC) migration is one of the most studied model systems for mesenchymal collective migration, in which cells are gathered into characteristic chains or streams within a physically restricted environment (*Szabó and Mayor, 2018*). In *Xenopus* embryos, cranial NC streams emerge from the interaction with neighboring tissue placode, where transient cell-cell interactions called contact inhibition of locomotion confer the supra-cellular polarity to determine the orientation of movement as a group (*Carmona-Fontaine et al., 2011*; *Hiraiwa, 2019*). Whereas the mechanism of this streaming migration is relatively well investigated (*Bronner and Simões-Costa, 2016*), it is still largely unknown how mesenchymal cells that are not tightly confined, such as those in mesoderm, move toward their destination and how their transient cell-cell adhesions contribute to it.

Mesoderm is a germ layer consisting of mesenchymal cells, and it forms during gastrulation. In the case of chick embryos, as the primitive streak is being formed, cells in the superficial layer (epiblast) move toward the primitive streak. Most of the mesoderm cells are formed by the convergence of these epithelial-shaped epiblast cells to the primitive streak, which subsequently undergoes the epithelial-to-mesenchymal transition (EMT). These mesoderm cells ingress adopting an irregular mesenchymal morphology and acquire high motility (*Nakaya et al., 2008b*; *Nakaya and Sheng, 2008a*; *Voiculescu et al., 2014*). Then, the mesoderm cells move away from the primitive streak at various anterior-posterior positions, in the three-dimensional (3D) space between the epiblast and endoderm (*Figure 1A*) (Video: https://www.sdbcore.org/object?ObjectID=358) (*Chuai et al., 2012*; *Iimura et al., 2007*; *Nájera and Weijer, 2020*; *Nakaya and Sheng, 2008a*; *Nakaya et al., 2008b*; *Psychoyos and Stern, 1996*; *Voiculescu et al., 2014*). Previous reports suggested that the mesoderm cells migrate at high density and their long-range migration pathway was controlled by a balance between chemo-repulsion mediated by FGF8 secreted from the primitive streak and chemo-attraction mediated by FGF4 secreted from the head-process and notochord (*Chuai and Weijer, 2009*; *Yang et al., 2002*). It is not clear, however, whether they are essentially solitary cells following the same cues while occasionally contacting each other or whether collective effects are essential for the mesoderm migration. If the latter is the case, questions arise as to what kind of cell-cell interactions give rise to the collective property, and what spatial structure emerges beyond the single cellular scale when cells migrate collectively.

Mesoderm migration have been studied mostly in fly, fish, frog, chick, and mouse embryos. However, spatiotemporal patterns of migration differ among the models. In zebrafish prechordal plate, cells exhibit collective movement towards the animal pole, whose directionality is maintained by contact between cells (*Dumortier et al., 2012*). In contrast, in mice, live imaging of whole embryo indicated that the mesoderm cells migrate individually rather than collective (*Ichikawa et al., 2013*). More recently, in mice, embryonic mesoderm cells demonstrate random aspects in their migration behavior and neighboring cells tend to migrate in parallel (*Saykali et al., 2019*). Leveraging the flat and easily accessible nature of the early chick embryo, researchers have conducted extensive and well-documented experimental studies over many years. The chick embryo has been particularly valuable in studying the properties of the mesoderm, including the migration pathways that depend on cell fate (*Psychoyos and Stern, 1996*). Live imaging techniques have also been employed to visualize the migratory routes of cells emerging from various positions along the primitive streak, spanning from anterior to posterior regions (*Yang et al., 2002*).

In this study, we examined how primitive-streak-derived mesenchymal cells, which are mostly mesoderm cells but a small fraction of definitive endoderm cells may be included, migrate during early chick gastrulation. Our quantitative data analysis, including large-scale cell tracking and topological data analysis (TDA) on high-resolution microscopy images showed that the cells migrate collectively, and they are connected to each other via cadherin-mediated cell-cell adhesion and form a meshwork structure that exhibits a continual and rapid reorganization. This dynamic meshwork structure was reproduced by a theoretical model, which suggests that the morphology of the cells, the strength of cell-cell adhesion, and the cell density are key parameters for the meshwork formation.

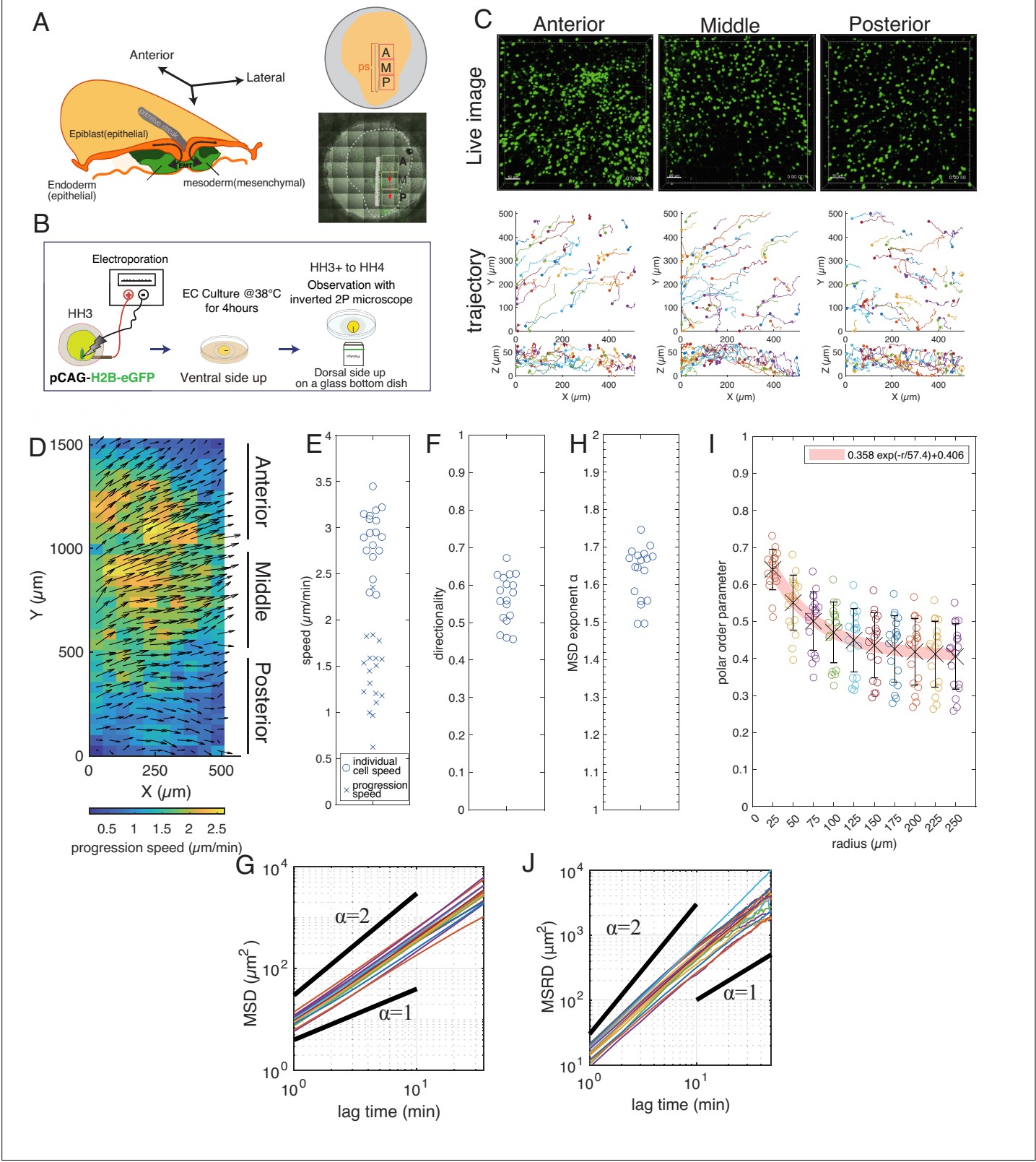

**Figure 1.** Mesoderm cells move collectively during gastrulation. (**A**) Schematic diagram of the chicken embryo at stage HH3. The observation regions are marked by the square boxes in the right panels (A: Anterior, M: Middle, P: Posterior). (**B**) Experimental procedure. DNA encoding H2B-eGFP was introduced into the cells in the primitive streak at stage HH3 by electroporation. After several hours of incubation, the position of the labeled nuclei was recorded using a multi-photon microscope. (**C**) Examples of the obtained images of the mesoderm cells expressing H2B-eGFP (upper panels)

*Figure 1 continued on next page*

*Figure 1 continued*

and reconstructed 3D trajectories (bottom panels). The x, y, and z axes correspond to the mediolateral, anterior-posterior, and dorsoventral axes, respectively. The initial position of the cells is marked by dots on the trajectories. Scale bar: 50 μm. (**D**) Spatial distribution of progression velocity (arrows) and progression speed (color). (**E**) Individual cell speed (o) and progression speed (x), and (**F**) directionality. Each data point of the individual cell speed and the directionality represents the average over the cells and that of the progression speed is the average over the subareas in each region of the six embryos. (**G**) Mean squared displacement (MSD). Each line corresponds to the MSD in each region of the 6 embryos. (**H**) Exponent of the MSD plotted in (**G**). (**I**) Polar order parameter $\varphi$ plotted against the radius of the measurement area. The polar order parameter calculated for the cells in the circular areas of a given radius at each time was averaged over the areas and time in each region of the 6 embryos (the crosses and error bars). (**J**) Mean squared relative distance (MSRD). Each line represents the MSRD in each region of the six embryos. On a time scale larger than about 10 min, the exponent of the MSRD becomes 1. The numbers of cells analyzed are N=1525 (A), 1112 (M), 791(P) (embryo 1), 371 (A), 416 (M), 235 (P) (embryo 2), 398 (A), 388 (M), 316 (P) (embryo 3), 230 (A), 296 (M), 175 (P) (embryo 4), 1386 (A), 1283 (M), 964 (P) (embryo 5), 1040 (A), 1102 (M), 496 (P) (embryo 6).

The online version of this article includes the following source data and figure supplement(s) for figure 1:

**Source data 1.** Trajectory data of mesoderm cells used in *Figure 1*.

**Figure supplement 1.** Localization and kinematic signatures of mesoderm cells.

## Results

### Mesoderm cells move collectively with frequent changes in their relative position

We first investigated how mesoderm cells migrate in the early gastrulating stage of chick embryos. To this end, we electroporated a plasmid encoding H2B-eGFP into the mesoderm precursor cells located at the primitive streak of stage HH3 embryos (*Hamburger and Hamilton, 1951*), and then the embryos were cultured ex vivo for several hours (*Chapman et al., 2001*). To follow the behavior of electroporated cells, we obtained 3D images (scan area: 500 μm × 500 μm, scan depth: 50 μm) of the mesoderm at three different streak levels along the primitive streak, at 1 min intervals, for 2 hr using an inverted two-photon microscope with 25 x WMP/0.25 NA objective (*Figure 1A and B*, and *Video 1*). We note that at this developmental stage (HH3), the entire length of the primitive streak is less than 2 mm. We also note that although the tissue scale extension along the anterior-posterior axis becomes more prominent after HH4+, the cell trajectories at this stage (HH3) reflect the cell migration in the environment of mesoderm region without the tissue-scale deformation.

From the position of the nuclei, we reconstructed the 3D trajectories of mesoderm cells (*Figure 1C* and *Video 1*), where the x and y axes correspond to the mediolateral and anterior-posterior (AP) axes, respectively. Most of these 3D trajectories were attributed to mesoderm cells (*Figure 1—figure supplement 1A*), although a small fraction of definitive-endoderm cells may also be present (*Psychoyos and Stern, 1996*). In all three regions, the mesoderm cells exhibited a spreading behavior with maximal cell displacement in particular directions (*Figure 1C*, Anterior, Middle, and *Figure 1D*). In the anterior and middle regions, the mesoderm cells tended to move in the anterior-lateral direction (*Figure 1C* Anterior, Middle and *Figure 1D*). In contrast, the mesoderm cells in the posterior region tended to move toward the lateral direction (*Figure 1C*, Posterior and *Figure 1D*). These migration directions were consistent with previous reports (*Nájera and Weijer, 2020*; *Yang et al., 2002*). Now we quantitatively investigate the migration of individual mesoderm cells and how their migration are coordinated among neighboring cells.

The trajectory of the individual cells showed that the mesoderm cells frequently changed their direction of motion, and some cells move even

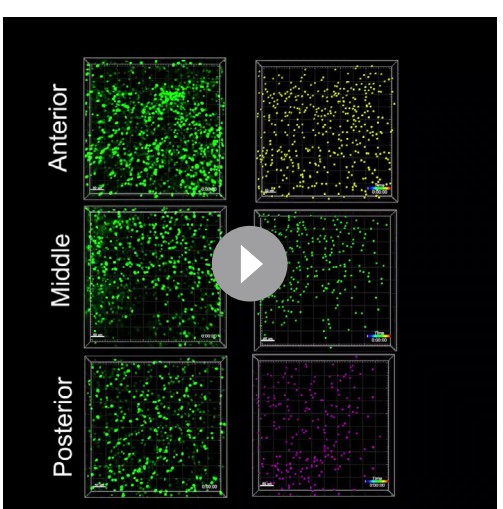

**Video 1.** Mesoderm cell movements on gastrulating chick embryo. Left: Nuclei of mesoderm cells are labeled by H2B-eGFP expression (green). Right: Cell trajectories by IMARIS tracking in the anterior, middle and posterior regions. Scale bars, 50 μm.

https://elifesciences.org/articles/84749/figures#video1

toward the primitive streak transiently (**Figure 1C**). The z-position of the mesoderm cells also changed frequently (**Figure 1C**). In fact, the distribution of the difference between the maximum and minimum z-positions of individual trajectories in 60 min showed a peak around 20 μm with a tail that extended more than 40 μm (**Figure 1—figure supplement 1B**), indicating that most cells change their z-position by more than one to two cell lengths, given that the typical cell length is 10–20 μm. Thus, the mesoderm cells frequently change their direction of motion in 3D.

We first quantitatively characterize this migration behavior of individual cells for trajectories obtained from six embryos (in total, 6 × 3 = 18 samples) (see Materials and methods). The mean values of individual cell speed for each sample were distributed from about 2.3 to $3.5 \mu m/min$ (**Figure 1E**) (for the speed of individual cell, see **Figure 1—figure supplement 1C**). The directionality defined by the ratio of the start-to-end distance to the total path length measures how persistent the trajectories are (Materials and methods). The directionality for the trajectories of 20 min length in 18 samples was smaller than one and was distributed from about 0.45–0.7 (**Figure 1F**). We then performed a mean squared displacement (MSD) analysis to evaluate the randomness of the individual mesoderm migration (Materials and methods). The MSDs were proportional to $t^{\alpha}$, where $\alpha$ was distributed from about 1.5–1.75 (**Figure 1G and H**). This result means that the mesoderm cell motion in all regions is in the regime between random walk ($\alpha = 1$) and ballistic motion ($\alpha = 2$) (**Gorelik and Gautreau, 2014**). We also confirmed that the MSD exponent and the directionality showed a good positive correlation (**Figure 1—figure supplement 1D**). To further verify the persistence of the migration direction, we calculated the autocorrelation function (ACF) of the velocity of individual cells (**Figure 1—figure supplement 1E**). The ACF indicated that the correlation decreased below 0.5 in 2 min for all 18 samples, indicating that the migration direction shows random fluctuations. For a longer time scale, the ACF gradually decreased, but did not converge to zero, indicating that the motion was biased slightly to a particular direction in each region. Thus, the motion of mesoderm cell is considered as a biased random walk. We note that these migration characteristics were not significantly different between the three regions: anterior, middle, and posterior.

We next elucidate the coordination of cell migrations among neighboring cells by analyzing the collective order of the mesoderm cell motions. To this end, we measured the polar order parameter $\varphi$ of the direction of cell motion that quantifies the instantaneous directional alignment among cells (Materials and methods) (**Méhes and Vicsek, 2014**). When all cells move in the same direction, $\varphi$ is one, whereas $\varphi$ is zero when the direction of the cell migration is completely random. First, we calculated $\varphi$ for cells within a circular region of radius 25 μm in the xy-coordinate. The value of $\varphi$ was distributed from 0.5 to 0.75 with a mean value of 0.64, indicating that the migration direction is well aligned among the neighboring cells (**Figure 1I**), despite of the randomness in the individual cell motion. To investigate how far the cell migration direction is aligned, we measured the polar order parameters for different radius of the measurement circle. As the radius of the measurement circle increases, the polar order parameter decreases exponentially with a characteristic decay length of 57 μm, beyond which the average of $\varphi$ is less than 0.5. This result indicates that the mesoderm cells within a length scale of about 60 μm migrate collectively. In the longer length scale, the migration of the mesoderm cells becomes less coordinated due to the fluctuation in the migration direction as the ACF of the velocity indicates.

To study whether the mesoderm cells within this characteristic length move together, we measured the mean squared relative distance (MSRD) (Materials and methods), which quantifies the temporal change of the relative distance between two cells. The MSRD curve increases with time, but there is a threshold time about 10 min (**Figure 1J**) beyond which the increase of the MSRD slows down. In this timescale, the two cells initially in contact move away from each other to about 25 μm. Beyond the threshold time, the MSRD increases linearly in time. This indicates that the migrating mesoderm cells do not form a tight cluster, changing their relative positions frequently. It also suggests that there is sufficient extracellular space where the cells can easily change their relative positions in the mesoderm tissue. We note that the Péclet number can be estimated to be 2–4 considering the mean cell speed obtained above, the typical cell size to be about 10 μm, and the persistent time obtained from the MSRD analysis.

We finally measured the progression speed of mesoderm for 18 samples, which was obtained as the time average of the average velocity of cells in small regions (50 μm × 50 μm) (**Figure 1D** color code). The mean of the progression speed was distributed from about 0.5–2 μm/min (**Figure 1E**),

which was almost the half of the individual cells speed (*Figure 1E*, *Figure 1—figure supplement 1C*). This difference can be explained by considering the randomness of cell migrations and the value of the polar order parameter of this length scale.

## Mesoderm cells form meshwork structure

If there is sufficient extracellular space for cells to change their relative position frequently, how do they distribute in 3D space between the epiblast and the endoderm? To explore this question, we prepared 3D reconstructed images of the mesoderm tissue at the mid-streak level in HH4 embryos (*Figure 2A*). From the projected image on the xy plane (*Figure 2B*, z - projection), we found no clear pattern in the distribution of cells in the primitive streak (dense region of nuclei stained with DAPI) and the mesoderm next to the streak, consistent with previous reports based on scanning electron microscopy (SEM) analysis (*England and Wakely, 1978*; *England and Wakely, 1977*). However, when we looked closely at the horizontal sections of 1.5 μm thickness, we realized that cells were not distributed uniformly but there were many holes without cells (*Figure 2B*, z-section, and *Video 2*). To see how these holes are formed, we visualized the cell-cell adhesion by staining for N-cadherin. It revealed that the mesoderm cells were connected via N-cadherin-mediated cell-cell adhesion and surrounded the holes, which led to formation of a meshwork structure within the tissue (*Figure 2B, C and D*, and *Video 2*). Thus, in contrast to previous reports (*Chuai et al., 2012*) that suggested that the mesoderm is densely packed with migrating cells, we found a meshwork structure of collectively migrating mesoderm cells. The holes in the meshwork structure were surrounded by about 10–20 cells (*Figure 2C*). The transverse section of the embryo revealed that the holes extended in the z-direction between the epiblast and the endoderm layers (*Figure 2—figure supplement 1A*).

To characterize the meshwork structure of the mesoderm quantitatively, we applied persistent homology analysis (Materials and methods). Persistent homology is a tool that has been developed recently in applied mathematics to quantitatively characterize the topological structure in disordered systems (*Buchet et al., 2018*; *Hiraoka et al., 2016*; *Obayashi et al., 2018*). To analyze an image by persistent homology, we first prepare a black and white binary pixel image from the original image by thresholding, where the pixels occupied by cells are white. Such a binary image can be seen as a landscape with hills and valleys, where hills represent cluster of cells, and the valleys represent holes between them. We can assign height and depth (negative height) of hills and valleys, respectively, by considering the shortest distance (Manhattan distance) from the interface between white and black pixels. Now, we consider this landscape at different heights. To uncover topological features, we apply progressive thresholding to the binary image from the minimum height (leading to completely white) to the maximum (leading to completely black) in a stepwise manner. At each threshold level, pixels below the threshold are black. As the threshold level incrementally changes, a feature, such as a hole which is a black area surrounded by a white area, emerges or 'is born'. Such a threshold level is recorded as 'birth level' of the feature. As the threshold level is increased further, the feature merges with another feature or 'dies'. Such a threshold level is recorded as 'death level' of the feature. Thus, the holes are characterized by the two quantities called birth and death levels, and they are visualized by points in persistence diagram (PD) where the coordinates are given by the birth and death levels (*Figure 2F*). These two quantities basically represent the size of the feature and the distance between two features, respectively. The difference between the death and the birth levels is called 'persistence', which becomes large for a reliable topological feature. Note that in the persistent homology on binary images, the birth and death levels, and thus the persistence, are measured in the unit of pixels.

We performed persistent homology analysis by using a software named HomCloud (*Obayashi et al., 2018*) and applying it to binarized images of the mesoderm at three different z levels. The PDs in *Figure 2F* show the results correspond to the images in *Figure 2E*. The points with large persistence were distributed around the death level ≈0 as a branch extended away from the diagonal line (*Figure 2F*, *Figure 2—figure supplement 1B*), which we call *hole branch* in this paper. Each point in this hole branch corresponded to a clear hole in the binary image (*Figure 2—figure supplement 1B*). To characterize the size of the holes quantitatively, we calculated the radius from the birth level of a hole in the hole branch (*Figure 2F*, see also Materials and methods), which was estimated on average as 17 μm, 15 μm, and 12 μm for the upper, middle, and lower layers, respectively (*Figure 2G*). Thus, the diameter of the hole is about half of the characteristic length scale of collective migration measured in the previous subsection (~60 μm). To conclude, these results indicate that the mesoderm

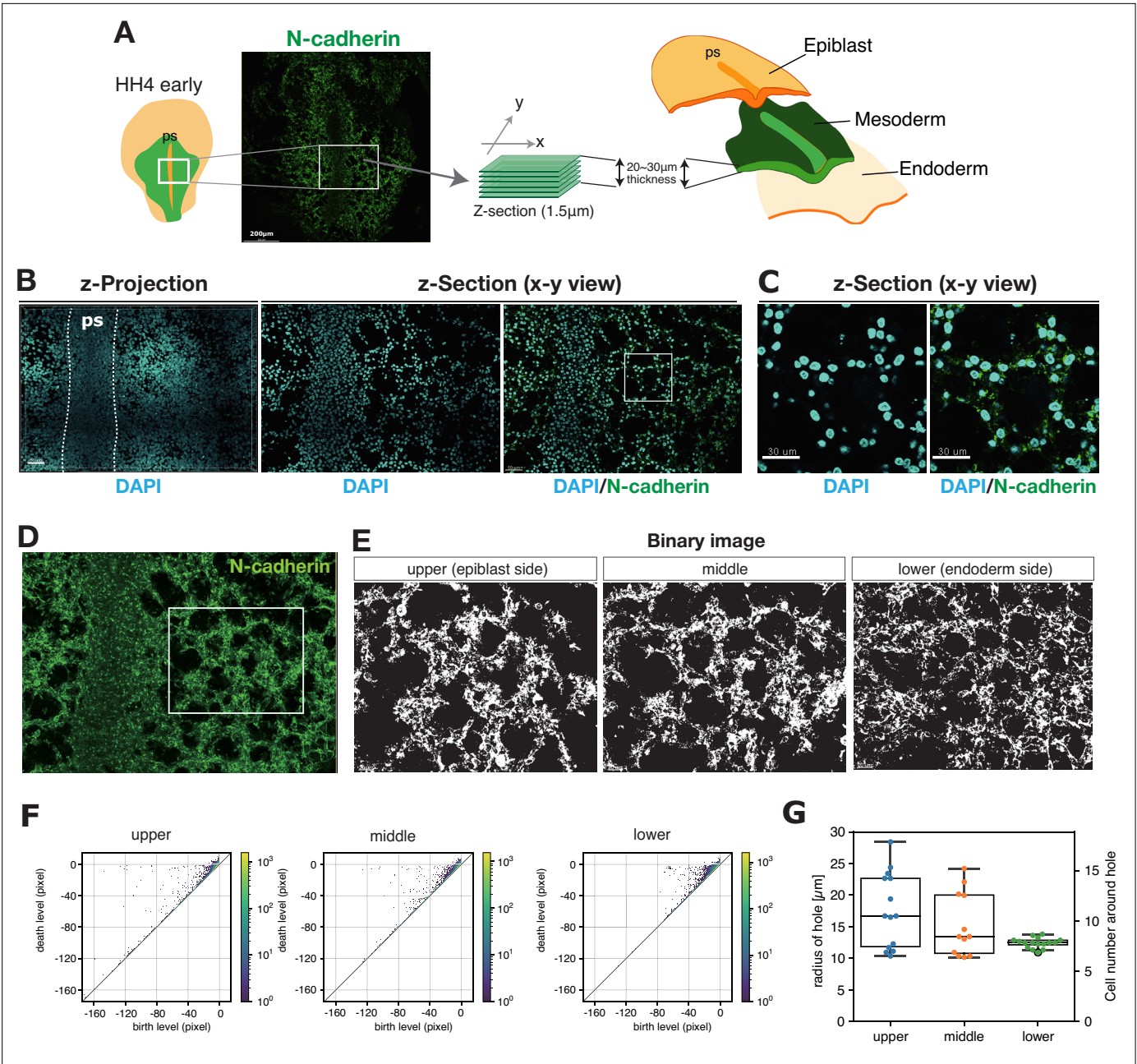

**Figure 2.** Meshwork structure in mesoderm during gastrulation. (**A**) Schematics of the 3D imaging. The white box indicates the imaged area shown in (**B**). (**B**) Spatial distribution of the cells in the fixed mesoderm tissue stained for nuclei (cyan) and N-cadherin (green) in the z-projection view (left) and the horizontal section (middle and right). (**C**) Magnified view of the characteristic meshwork structure in the white box in (**B**). (**D**) N-cadherin expression in the middle section of the mesoderm. (**E**) Binary images of three z-sections in the white box in (**D**). (**F**) Persistence diagram (PD) obtained by applying persistent homology analysis to the three z-sections in (**E**). The pixel size in (**E**) is 0.192 μm. The points forming a hole branch around the death level ~0 correspond to the holes. Holes are identified as the points with a birth level smaller than –10 μm and a death level larger than –2.5 μm (see Method). The color indicates the multiplicity of the points. (**G**) Statistics of the radius of holes that appear in the hole branch in the PD and the number of the cells surrounding the holes given that the cell diameter is 10 μm.

The online version of this article includes the following source data and figure supplement(s) for figure 2:

**Source data 1.** Numerical data plotted in *Figure 2G*.

**Figure supplement 1.** Three-dimensional visualization of holes and their detection by persistent homology.

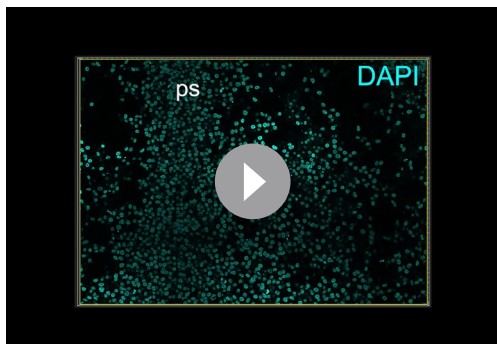

**Video 2.** Meshwork structure in the mesoderm. Confocal Z-stack images of the mesoderm and the primitive streak of stage HH4 chick embryo. The embryo is stained for nuclei with DAPI (cyan) and for N-Cadherin (green). Z-stack images with a thickness of 1.5 μm show that the characteristic meshwork structures are composed of multiple cells in the mesoderm located on both sides of the primitive streak. ps, primitive streak.

https://elifesciences.org/articles/84749/figures#video2

is extended as a loose layer of cells, and the characteristic meshwork structure is formed by cells connected by N-cadherin-mediated cell adhesion, which may lead to their collective migration.

## Meshwork structure is dynamic with the emergence and collapse of holes

To examine whether the meshwork structure was also observed in living embryos, we next performed live imaging of transgenic chicken embryos that ubiquitously express cytoplasmic GFP (*Motono et al., 2010*) to visualize the migration of the mesoderm cells and how they interact with each other (*Figure 3A*). We again found that cells away from the primitive streak formed meshwork structures, as seen in the optical thin section of middle layer in the mesoderm at the mid-streak level (*Figure 3A*, *Figure 3—figure supplement 1A*, , and *Video 3* and *Video 4*). Interestingly, the holes in the meshwork were not static but gradually moving anterior-laterally (*Figure 3A*, *Figure 3—figure supplement 1B, C*, and *Videos 3 and 4*). We also notice that a cell of one hole migrates and participates in another hole as time passes. (*Figure 3A* and *Videos 3 and 4*).

To confirm this dynamic meshwork structure is the same structure as those found in the fixed embryos in the previous subsection, we applied persistent homology analysis to the snapshots of the time-lapse images (*Figure 3A*). The PDs at different time points (0, 12, and 24 min) showed the hole branch of the points with large persistence, which was comparable to the PDs obtained for the fixed embryo (*Figures 2F and 3B*). From this, we conclude that the meshwork structure found in the living embryo is the same structure as those found in the fixed embryos. The radius of the holes was on average about 13 μm, which showed no systematic change during the observation over 24 min. (*Figure 3C*).

Finally, we investigate how the meshwork structure changes in time. The advanced inverse analysis enables us to detect the region of the hole in the original binary image, which corresponds to each point in PD (Materials and methods and *Figure 3—figure supplement 1D*). To visualize the time evolution of the holes, we plotted the region of some holes obtained by the advanced inverse analysis in x-y-t coordinates over 24 min (*Figure 3D*). The holes underwent emergence and collapse, and they also split and merged occasionally, while moving gradually in the anterior-lateral direction (*Figure 3D*, *Figure 3—figure supplement 1B, C*).

These results implied that the mesoderm cells are only transiently connected to each other and can easily change their partners. Thus, the meshwork structure is dynamic in the sense that cells that form a hole are replaced over time while the mesoderm cells migrate away from the primitive streak. Since the characteristic length scale of collective migration and the diameter of the holes are comparable, the cells that form a hole move in roughly the same direction. However, frequent changes in the relative positions among the cells cause the collective order to decay over long length scales much larger than the hole size (*Figure 1I*).

## N-cadherin appears to play an important role for collective cell migration during mesoderm formation

Previous studies based on scanning and transmission electron microscopy observations have reported that the space between the epiblast and endoderm in early chick embryos, where the mesoderm cells migrate, is filled with water-soluble components such as hydrated glycosaminoglycans, in particular hyaluronic acid, and contains little extracellular matrix that can provide a scaffold for cell migration (*Van Hoof et al., 1986*; *Sanders, 1986*; *Sanders, 1979*). This suggests that the mesoderm cells rely

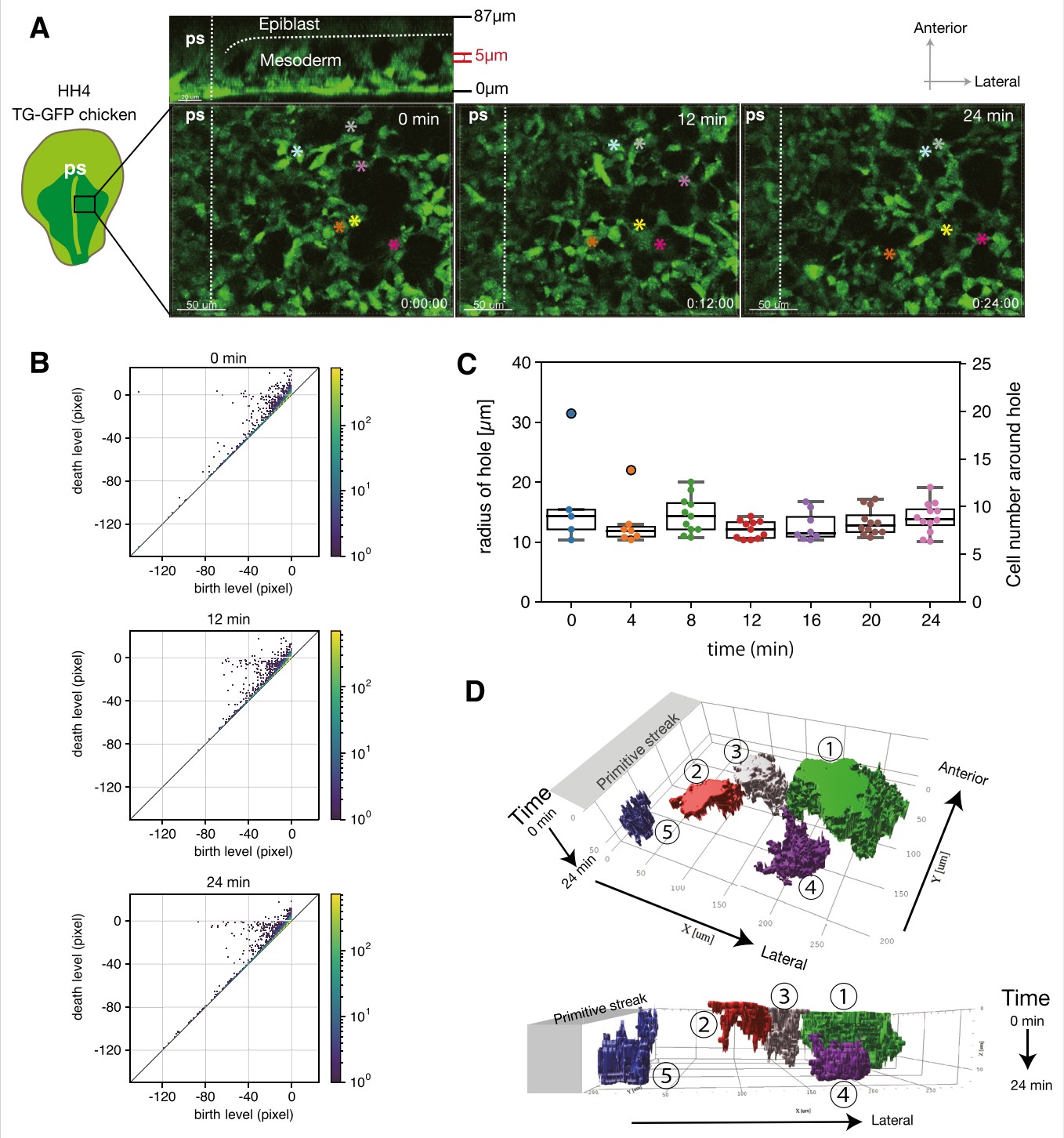

**Figure 3.** Dynamic meshwork structure. (**A**) Successive snapshots obtained from a live image of mesoderm tissue. The position of the six cells at different time points are indicated by the colored asterisk. (**B**) Persistence diagram (PD) of the three snapshots in (**A**). The hole branch of the points around death level ~0 away from the diagonal line. The pixel size in (**A**) is 0.22 µm. Holes are identified as the points with a birth level smaller than –10 µm and a death level larger than –2.5 µm (see Method). The color indicates the multiplicity of the points. (**C**) The time series of the radius of holes that appear in the hole branch in the PD, and the corresponding number of the cells surrounding the holes that is calculated from the radius under the assumption that the cell diameter is 10 µm. The p-values between any two time points obtained by t-test were larger than 0.05 except for the pairs of 0 min and 8 min, 0 min and 12 min, 0 min and 16 min, 0 min and 20 min, 0 min and 24 min, 4 min and 12 min (p<0.01), and 8 min and 12 min (p<0.05), which might possibly be caused by the small size of the data set. (**D**) Spatiotemporal diagram of the holes. The holes were dynamic with the appearance (4) and disappearance (2,3), as well as the fusion (5) and fission (2).

*Figure 3 continued on next page*

*Figure 3 continued*
The online version of this article includes the following source data and figure supplement(s) for figure 3:
**Source data 1.** Numerical data plotted in *Figure 3C*.
**Figure supplement 1.** Dynamics of holes.

on cell-cell adhesion to get traction to migrate, as well as the contact to the basal lamina of either epiblast or endoderm (*Brown and Sanders, 1991*; *Sanders, 1986*). The formation of the meshwork and its dynamic characteristics may also rely on cell-cell adhesion. Taking these points into consideration, we questioned how much impact cell-cell adhesion has on cell migration as well as on the meshwork structure.

Consistent with the previous reports (*Moly et al., 2016*; *Nakaya et al., 2008b*; *Yang et al., 2008*), during gastrulation, most mesoderm cells expressed the classical adhesion molecule N-cadherin (*Figure 2A*). Notably, we found that N-cadherin was localized at the cell-cell contact site between the cells in both the x-y and x-z sections (*Figure 4A*), meaning that the N-cadherin-mediated cell-cell adhesion is present at the cell-cell contact sites between upper and lower cells as well as at the horizontal cell-cell contact site. This implies that N-cadherin-mediated cell-cell adhesion plays a fundamental role in the formation of the meshwork structure and the collective migration. To test this idea, we studied the effect of reducing the intercellular adhesion of mesoderm cells (*Nieman et al., 1999*; *Ozawa, 2015*; *Ozawa and Kobayashi, 2014*). To this end, we generated a deletion mutant of N-cadherin (N-Cad-M) which lacks the extracellular (EC) domain that is responsible for adhesive activity (*Nieman et al., 1999*; *Ozawa, 2015*; *Ozawa and Kobayashi, 2014*). In addition, H2B-mCherry was flanked on its 3'-side of the 2 A peptide to make the N-Cad-M expressing cells detectable (*Figure 4B*). We over-expressed N-Cad-M with H2B-mCherry in the mesoderm cells.

To ensure that the endogenous N-cadherin was disappeared from the membrane, we used an N-cadherin antibody reactive against the EC domain, which can detect only endogenous N-cadherin. Immunostaining with this antibody showed that the endogenous N-cadherin accumulated in the cytoplasm and its expression was disappeared from the membrane of the mutant mesoderm cells (*Figure 4C* left, cells with nuclei labeled in red). Similarly, endogenous expression of P-cadherin was affected, which was also mainly detected in the cytoplasm of mutant cells (*Figure 4C* right, cells with nuclei labeled in red). From these results, we suppose that the preferential localization of large amounts of N-Cad-M to the plasma membrane disrupted the membrane localization of endogenous cadherins, which effectively attenuates cadherin-mediated cell-cell adhesion. Indeed, the cells expressing N-Cad-M tend not to be integrated into the meshwork of the control cells (*Figure 4D*). In addition, the mutant

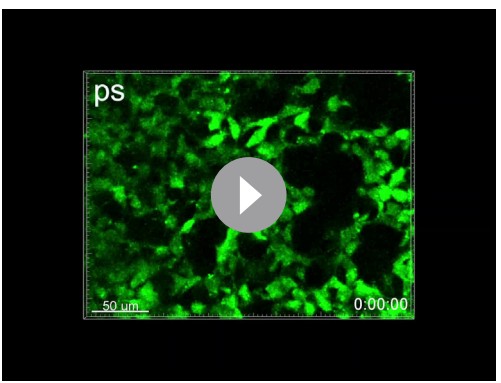

**Video 3.** Mesoderm cells forming a dynamic meshwork during migration. Live imaging of a thin section (5 μm) of the mesoderm of stage HH4 GFP-expressing transgenic chicken embryo. The mesoderm cells migrate from the primitive streak by forming a dynamic meshwork structure undergoing continual and rapid reorganization. ps, primitive streak. Scale bar, 50 μm.
https://elifesciences.org/articles/84749/figures#video3

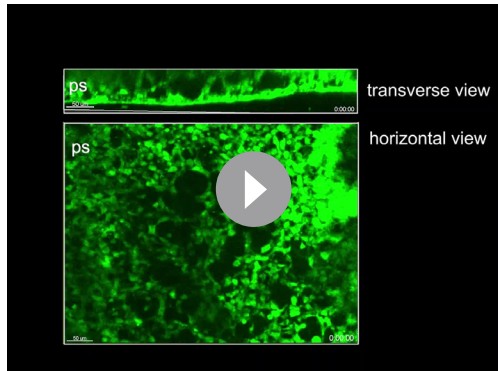

**Video 4.** Dynamics of the meshwork structure. 4D live imaging of the mesoderm of stage HH4 GFP expressing transgenic chicken embryo. The optical transverse section and horizontal section, monitored for 40 min, shows that the three-dimensional dynamic meshwork structure is formed by the migrating mesoderm cells and the holes move toward the anterior-lateral direction. ps, primitive streak. Scale bar, 50 μm.
https://elifesciences.org/articles/84749/figures#video4

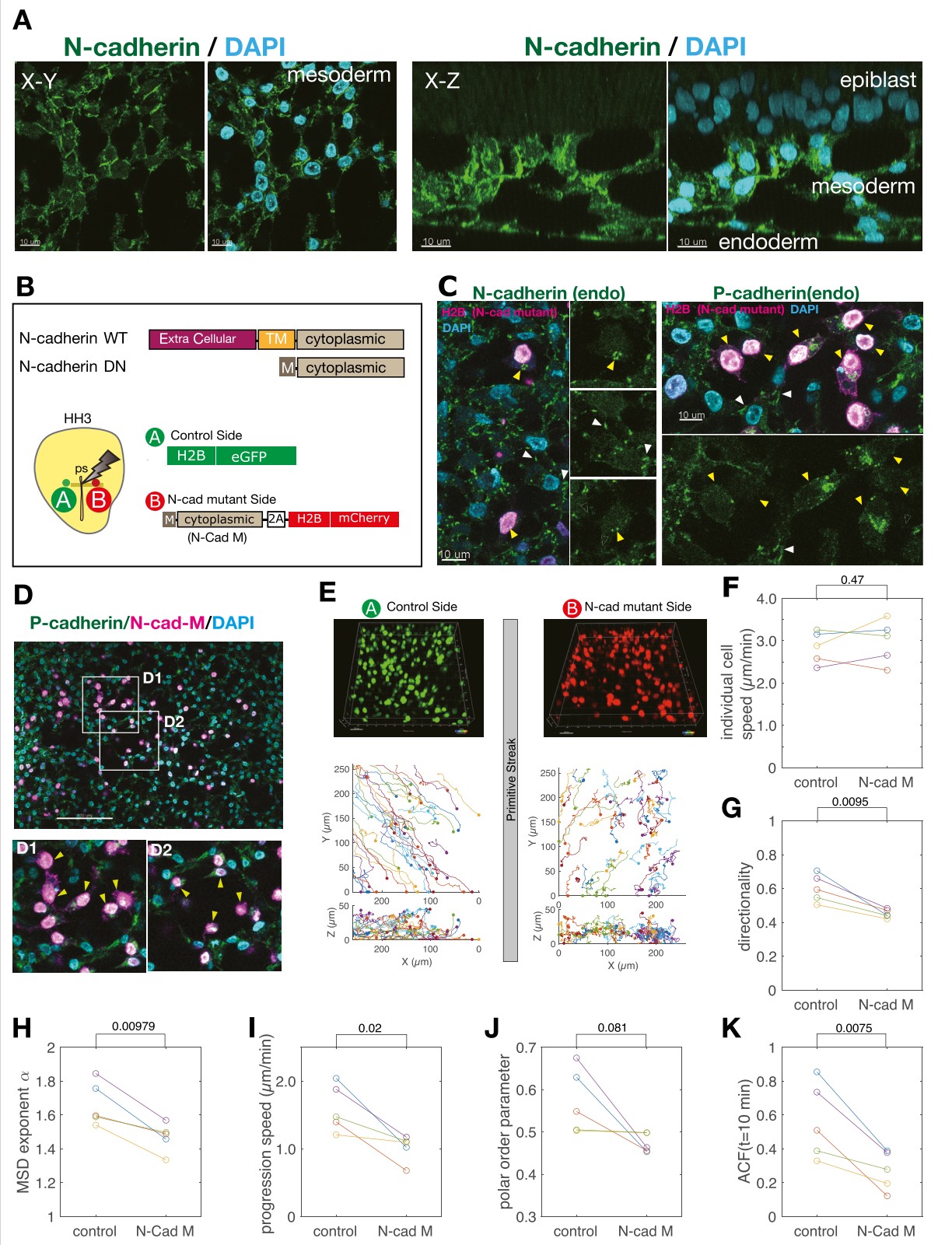

**Figure 4.** Intercellular adhesion controlling collective mesoderm cell migration. (**A**) N-cadherin expression in the mesoderm. N-cadherin was localized at the cell-cell contact sites both in the horizontal section (left) and in the vertical section (right) that surround the holes. Scale bars, 10 μm. (**B**) Structure of the wild-type N-cadherin and the deletion mutant of N-cadherin consisting of the cytoplasmic domain with myristoylation signal (top). Schematic diagram of the experimental method (bottom). To compare the migration of the mesoderm cells, H2B-eGFP was electroporated on the A side, while the

*Figure 4 continued*

N-cadherin mutant (N-Cad-M) was electroporated on the B side. The N-Cad-M expressing cells were marked by the H2B-mCherry expression. (**C**) Effects of N-Cad-M overexpression on endogenous cadherin expression. Endogenous N-cadherin (left) and P-cadherin (right) are expressed specifically at the cell-cell contact site in the control mesoderm cells (white arrow heads). In contrast, in the cells expressing N-Cad-M labeled in red, the expression of N-cadherin (left) and P-cadherin (right) were almost disappeared from the cell membrane (yellow arrow heads). Scale bars, 10 μm. (**D**) N-Cad-M expressing cells tend not to be integrated into the meshwork structure of control mesoderm cells (yellow arrow heads). (**D1**) and (**D2**) Magnified images in the white boxes in the top panel. The N-cadherin (N-Cad-M) expressing cells did not participate in the meshwork. Scale bar, 100 μm. (**E**) Examples of (top) a snapshot of the live imaging and (bottom) trajectories of the mesoderm cells expressing H2B-eGFP (A side) and the N-Cad-M (B side) of the same embryo. The initial position of the cells is marked by dots on the trajectories. (**F–K**) Statistical quantification of the migration behavior of the control and N-Cad-M cells for five embryos. The corresponding statistical quantity of each cell in each embryo is shown in *Figure 4—figure supplement 2*. The quantities of control and N-Cad-M in the same embryo are linked by the line. (**F**) Mean of individual cell speed (p=0.47). (**G**) Mean of directionality (p=0.0095). (**H**) MSD exponent (p=0.00979). (**I**) Mean of progression speed (p=0.02). (**J**) Polar order parameter (p=0.081). (**K**) Auto-correlation function (ACF) of the direction of collective migration at 10 min (p=0.0075). *p*-values were obtained using paired t-test of the five embryos. The xy size of the imaged square area of five embryos: 258, 192, 207, 500, 500 μm. The numbers of cells analyzed: N=118 (Control), 119 (Mutant) (embryo 1), 41 (C), 28 (M) (embryo 2), 44 (C), 30 (M) (embryo 3), 253 (C), 290 (M) (embryo 4), 297 (C), 232 (M) (embryo 5).

The online version of this article includes the following source data and figure supplement(s) for figure 4:

**Source data 1.** Trajectory data of mesoderm cells used in *Figure 4*.

**Figure supplement 1.** Multiphoton images of the control cells and the N-cadherin (N-Cad-M) expressing cells ('Raw image' in the left panel).

**Figure supplement 2.** Statistical comparison between control and N-Cadherin mutant expressing cells within individual embryos.

**Figure supplement 3.** Cell-cell contact behaviors in control and N-Cadherin mutant expressing cells.

cells were less elongated and tended to exhibit a more rounded shape (*Figure 4—figure supplement 1*, see also *Video 5*).

Using this mutant form of N-cadherin, we performed live imaging to investigate the effect of the cell-cell adhesion on the mesoderm cell migration. To compare with the control cells, we electroporated the N-Cad-M construct into the mesoderm cells on one side of the primitive streak, and we introduced a plasmid expressing H2B-eGFP to the cells on the other side to trace them as control cells (*Figure 4B and E*, *Video 6*). We performed

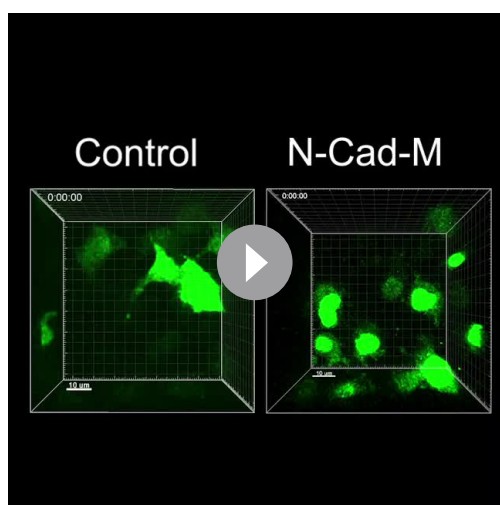

**Video 5.** Cell-cell contact behaviors in the N-Cadherin mutant expressing cells and the control cells. Cell membrane of the control cells and of the N-Cad-M expressing cells are detected by GFP-CAAX expression. Left: The control cells undergo continual contact with the surrounding cells. Right: The N-Cad-M expressing cells change the contact partners one after another during the observation. Scale bars, 10 μm.
https://elifesciences.org/articles/84749/figures#video5

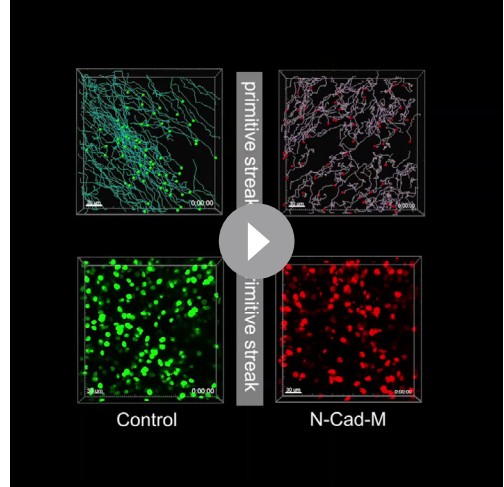

**Video 6.** Cell trajectories of the N-Cadherin mutant expressing cells and the control cells. Left, Control side: Nuclei of mesoderm cells are labelled by H2B-eGFP expression (control, green). Most mesoderm cells away from the primitive streak migrate toward the anterior-lateral direction. Right, N-cadherin mutant side: The N-cadherin deletion mutant expressing cells are detectable by H2B-mCherry expression. These cells also migrate in the anterior-lateral direction but exhibit zigzag trajectories, which is apparently different from the control cells. Scale bars, 30 μm.
https://elifesciences.org/articles/84749/figures#video6

this analysis for five embryos, each of which contains more than 60 cells (see *Figure 4E-K*, *Figure 4—figure supplement 2*). We found that the cells expressing N-cadherin mutant exhibited meandrous motion with more frequent changes in migration direction than the control cells (*Figure 4E*, *Video 6*), although the migration speed along the trajectory was not statistically different between them (*Figure 4F*, *Figure 4—figure supplement 2A*). This is confirmed by the significant reduction of the directionality (*Figure 4G*, *Figure 4—figure supplement 2B*) and the smaller exponent of the MSD for the mutant cells (*Figure 4H*). In addition, the progression speed of mutant cells was lower than that of the control cells for each embryo (N=5) (*Figure 4I*, *Figure 4—figure supplement 2C*), indicating that the progression of mesoderm tissue depends on N-cadherin. These results show that N-cadherin appears to play an important role in the tissue progression and the directionality of the mesoderm cell migration.

We next investigate the role of N-cadherin on the collective migration of the mesoderm cells. While the polar order parameter $\varphi(t)$ on the mutant side tended to be smaller than that on the control side, the overall difference was not statistically significant (*Figure 4J*, *Figure 4—figure supplement 2D*). To see the persistence of the direction of collective migration, $P(t)$, we measured its auto-correlation function (ACF) (Materials and methods). The ACF of the direction of collective migration for the mutant cells decayed faster than that of the control cells (*Figure 4K*, *Figure 4—figure supplement 2E*), indicating that the mutant cells changed the direction of collective migration more frequently than the control cells did. These results indicate that the N-cadherin can contribute to maintain the persistence of the direction of collective migration.

To explore potential differences in intercellular contact dynamics between control and mutant cells, we carefully examined time-lapse images. Qualitative observations suggest that control cells tend to elongate their bodies and establish prolonged cell-cell contacts via protrusions (*Figure 4—figure supplement 3*). These cells remained in contact for several tens of minutes, with the longest contact duration exceeding 1 hr (No.1 and No.2 pair in *Figure 4—figure supplement 3*, upper panels, *Video 5*). In contrast, mutant cells appeared to exhibit shorter-lived interactions, as they typically did not maintain cell-cell contacts for more than 20 min after colliding with neighboring cells (See No.1 cell in *Figure 4—figure supplement 3*, bottom panels, *Video 5*). While these observations suggest that reduced contact duration may contribute to more randomized movement of mutant cells, leading to frequent changes in the direction of collective migration, a more extensive quantitative analysis would be necessary to confirm this trend.

Taken together, the comparison of the migration characteristics between the control and N-cadherin mutant cells indicated that the N-cadherin appears to play an important role on the directionality of the individual mesoderm cells, their collective migration, and the tissue progression speed of the mesoderm (*Figure 4F–K*). We also found that the mutant cells tended to be more rounded than the control cells (*Figure 4—figure supplement 1*) and tend not to be integrated into the meshwork structure formed by the control cells (*Figure 4D*). We, therefore, hypothesize that cell-cell adhesion mediated by N-cadherin is one of the key factors for the formation of the meshwork structure. Unfortunately, for technical reasons, it was difficult to introduce N-Cad-M into all mesoderm cells to see if tissues composed only of N-Cad-M cells fail to form a meshwork structure. Therefore, we next tested our hypothesis by developing an agent-based theoretical model.

## Theoretical model to investigate the formation of meshwork structure

From the experiment using the mutant form of N-cadherin, we hypothesized that the cell-cell adhesion is one of the key factors for the meshwork structure formation. To understand how the dynamic meshwork structure of mesoderm cells emerges and how it is influenced by the cellular and intercellular properties, we develop an agent-based theoretical model. To this end, we modeled a cell by a rod-shape particle that interacts with others by short-range attraction with a repulsive core (Materials and methods). To focus on the essential aspect of the meshwork formation without complication, we will consider a model in two-dimensional space. We note that previous theoretical studies reported that elongated cells with attractive interaction can reproduce the formation of angiogenetic network structure (*Palm and Merks, 2013*; *Szabo et al., 2007*).

To start with, we studied the steady-state spatial distribution of agents, starting from the initial state where the agents were randomly positioned in space with random orientation. We investigate the impact of the cell-cell adhesion by changing the strength of the attractive interaction. To highlight

the role of the attractive interaction, we kept the aspect ratio $\gamma = 2$ and omitted the self-propulsion of the agents. When the attractive interaction was absent or small, the agents were distributed randomly without any clear spatial pattern (**Figure 5A**, $\epsilon_{atr} = 0, 0.001$). However, for a sufficiently large attractive interaction, the agents formed a meshwork structure (**Figure 5A**, $\epsilon_{atr} = 0.01, 0.05$). The birth level calculated by the persistent homology analysis, which corresponds to the radius of holes, increased with the strength of attractive interaction (**Figure 5B**). These results demonstrate that the cell-cell adhesion plays an important role in the meshwork structure formation.

Since the wild-type cells tended to be more elongated than the mutant cells (**Figure 4C and D**, for quantification see **Figure 4—figure supplement 1**), we next studied the impact of the aspect ratio for sufficiently large attractive interaction, i.e., $\epsilon_{atr} = 0.01$. When the aspect ratio $r$ was small, the agents were distributed randomly without any spatial pattern (**Figure 5C** $r = 1.5, 1.75$). However, there was a threshold aspect ratio $r^*(1.75 < r^* < 2)$, beyond which the agents formed a meshwork structure with many holes void of agents (**Figure 5C** $\gamma = 2, 2.25$). The persistent homology analysis distinguished this difference; at the threshold aspect ratio $r^* \approx 1.9$, the birth level of the holes, which corresponds to the radius of hole, showed a sharp increase (**Figure 5D**). This transition is evidence that the elongated shape with a large aspect ratio is important for the meshwork structure formation.

In summary, our in silico results confirmed that both attractive interaction and elongated shape with a large aspect ratio are necessary for the formation of meshwork structure. Indeed, the aspect ratio of mesoderm cells was 2.34±0.08 (±SEM) (**Figure 4—figure supplement 1**, control), while that of the N-cadherin mutant cells was 1.91±0.08 (±SEM) (**Figure 4—figure supplement 1**, N-Cad-M). We emphasize that this experimental result is consistent with the existence of the threshold aspect ratio at $r^* \approx 1.9$ in our in silico result (**Figure 5D**).

## Mechanism of meshwork formation

To understand the mechanism of the formation of meshwork structure, we focused on the small aggregates of agents that were found when the agent density was low. Since the results were quantitatively clearer for a high aspect ratio, we first set the agent aspect ratio as $r = 4$. When the density of agent is low, many small aggregates are formed due to the short-range attractive interaction (**Figure 5E** $\rho = 0.25$). Most of the aggregates have an elongated shape with an aspect ratio much greater than one (**Figure 5F** left, color). Inside the aggregates, the agents tend to align their direction of the shape elongation. Such a directional order is called nematic order. The direction of the nematic order in each aggregate is correlated with that of the aggregate elongation (**Figure 5F** left and the red curve in **Figure 5F** right), indicating that the aggregates tend to elongate in the direction of the nematic order. When the agent density increases beyond a threshold value, a meshwork structure is formed (**Figure 5E** $\rho = 0.6$). Thus, a scenario for the formation of the meshwork structure is as follows. The attractive interaction between agents induces the formation of aggregates, in which the agents align their orientation nematically. The positional and directional fluctuations of the agents deform the shape of aggregates in a way that they elongate in the direction of the nematic order. As the agent density increases, such elongated aggregates further extend and are eventually connected to each other, leading to the formation of the meshwork structure due to the randomness of the aggregate elongation direction. When the aspect ratio $r$ is reduced to $r = 2$, the correlation of the orientations of the nematic order and the aggregate elongation decreases (the blue curve in **Figure 5F** right). As the agent density increases, however, the correlation increases, indicating that the same scenario applies to this case ($r = 2$). In contrast, when $r = 1.75 < r^*$, the correlation decreases as agent density increases, suggesting that the scenario does not hold, resulting in the random distribution of agents without the meshwork structure formation.

## Dynamic meshwork formation with the supply of agents

During gastrulation, mesoderm cells are continuously supplied and move away from the primitive streak (**Figure 1A**). To mimic this situation, we modified the simulation condition to the case where the agents are supplied constantly at the rate from one side of the boundary, which corresponds to the primitive streak (Materials and methods). We also switched on the self-propulsion of the agents in the direction of the shape elongation and introduced a slight chemotaxis so that the agents move efficiently away from the primitive streak boundary (PS boundary). When the supply rate was high, the space was filled by the agents leaving no clear holes (**Figure 5G**, $r_{source} = 0.00019$). In contrast,

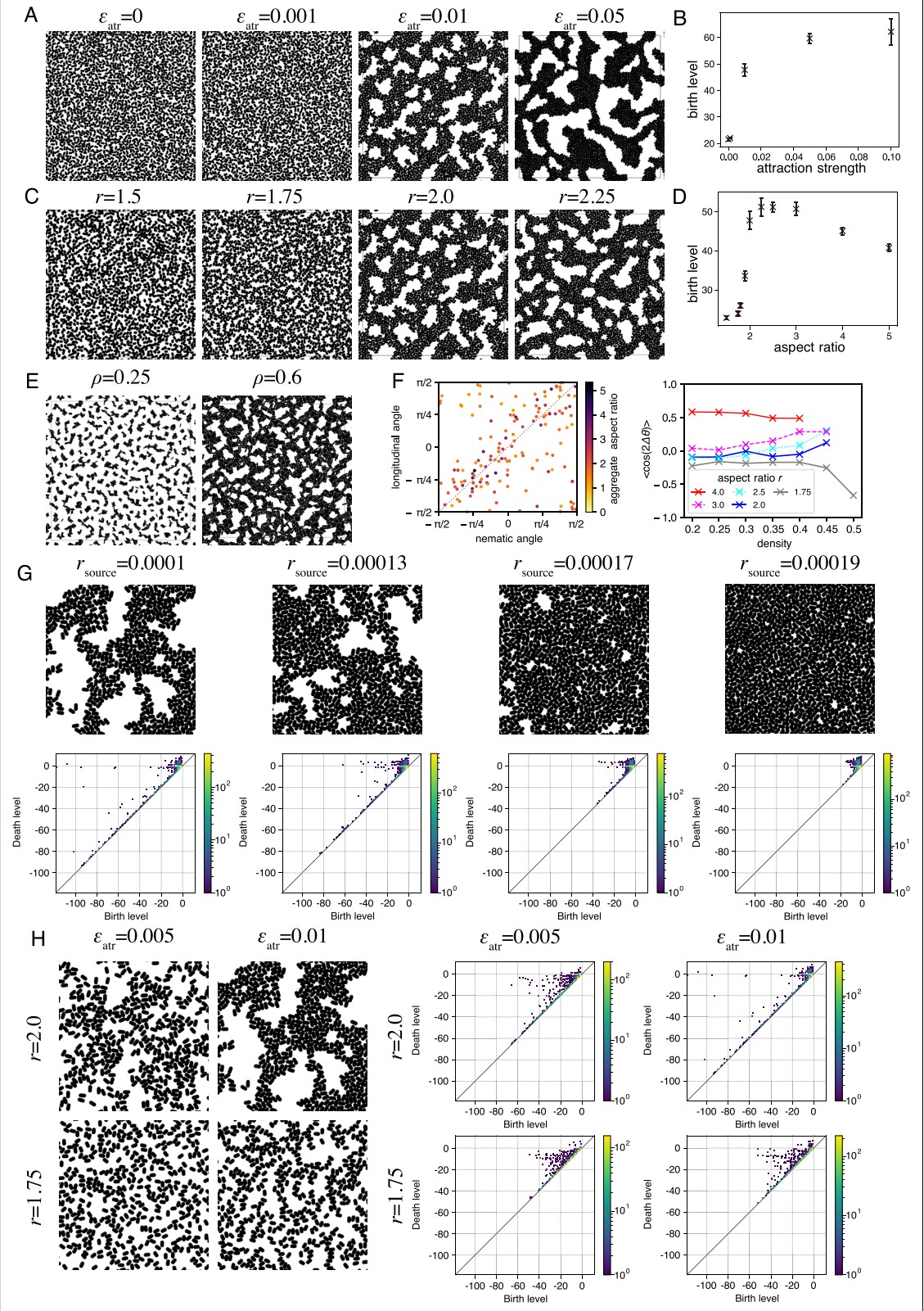

**Figure 5.** Theoretical model of meshwork formation. (**A**) Impact of the attractive interaction strength $\epsilon_{atr}$ on the meshwork structure formation. (**B**) Dependence of the birth level of the holes on the attractive interaction strength $\epsilon_{atr}$. The error bars indicate the standard error of mean obtained from n=10 independent simulations. (**C**) Impact of the agent aspect ratio $r$ on the meshwork structure formation. (**D**) Dependence of the birth level of the holes on the aspect ratio $r$ obtained from the persistent homology analysis. The error bars indicate the standard error of mean obtained from

*Figure 5 continued on next page*

*Figure 5 continued*

n=10 independent simulations. (**E**) Alignment of the agents in the aggregates and the meshwork structure. Aspect ratio $r$ = 4. (**F**) Relation between the nematic direction of agents in the aggregates and the elongation direction of the aggregates. (left) Relationship between the nematic angle and the longitudinal angle of each aggregate. The color of the points represents the aspect ratio of the aggregates. (right) The correlation between the nematic angle $\theta_n$ and the aggregate longitudinal angle $\theta_a$ defined by $\langle \cos 2 \left( \theta_n - \theta_a \right) \rangle$ as a function of the aggregate density for different aspect ratio $r$. (**G**) Impact of the agent supply rate on the meshwork structure formation in the simulation with the agents supplied from the PS boundary on the left. Snapshots (top) and persistence diagrams (PD) (bottom). (**H**) Impact of the adhesion and the aspect ratio on the meshwork structure formation in the simulation with the agents supplied from the PS boundary. Snapshots (left) and PD (right).

The online version of this article includes the following figure supplement(s) for figure 5:

**Figure supplement 1.** Theoretical model for the meshwork formation of chick mesoderm cells.

---

when the rate was decreased, small holes appeared, the size of which increased as the supply rate was further decreased (*Figure 5G*). These holes move away from the PS boundary in the lateral direction. The holes also showed dynamic behaviors, such as emergence, collapse, splitting, and merging, due to the agent self-propulsion (*Figure 5G*, $r_{source}$ = 0.0001, and *Video 7*), which resembled the experimental observations of TG-chick embryos (*Figure 3A*, *Figure 3—figure supplement 1B, C*, and *Videos 3 and 4*). To characterize the structure quantitatively, we performed persistent homology analysis and found that the points of birth-death pairs with large persistence were distributed in a hole branch around the death level ~0 when the supply rate was small ($r_{source}$ = 0.0001, *Figure 5G*, bottom), consistent with the experimental observation (*Figures 2F and 3B*). As the supply rate increases, the holes void of cells became smaller and, correspondingly, the hole branch shrank. In consequence, when the supply rate $r_{source}$ = 0.00019, the points were distributed in a clumped pattern near the diagonal line (*Figure 5G*, bottom). These results indicate that the supply rate, which controls the density of the cells in the mesoderm, is an additional important parameter for the meshwork formation. Note that the agents cannot form a meshwork structure at a very low density (*Figure 5E*). We also confirmed that the decrease in either the aspect ratio of agent shape or the attractive interaction prevented the formation of the meshwork structure when supply rate was $r_{source}$ = 0.0001 (*Figure 5H*).

## Dependence of mesoderm meshwork structure on the developmental stage

Now a new question arises: How do the meshwork structures of the mesoderm cells change during the embryonic development? To answer this question, we performed a persistent homology analysis using the horizontal slice images at different developmental stages. Interestingly, we found that as the developmental stage proceeds, the size of holes decreases and eventually the space was filled by cells (*Figure 6A*, top). The corresponding PDs showed that the points of birth-death pair with large persistence were distributed in a hole branch around death level ~0 at HH3+. However, as the developmental stage proceeded to HH4+, the hole branch shrank, and the distribution of the points was eventually changed to a clumped pattern near the diagonal line (*Figure 6A*, bottom). The average radius of holes was reduced from about 8 µm at HH3 + to 5 µm at HH4+, which took about 6 hr (*Figure 6C*). Note that the average radius of the holes at HH3 + becomes 15 µm when we focused on the larger holes by setting the threshold birth level to the same as that for *Figures 2G and 3C* (see also Materials and methods). Thus, while the size of holes is maintained for about half an hour (*Figure 3C*), it gradually decreases over several hours. From the simulation results shown in *Figure 5G*, we speculated that the supply rate of mesoderm cell from the primitive streak increases gradually as the developmental stage proceeds. Therefore, we performed a simulation with a time-dependent supply rate of the agent from the PS boundary (Materials and methods). We found

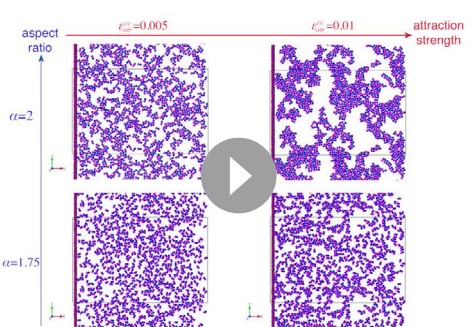

**Video 7.** Meshwork structure formation in the simulation with agent supply. The agents were supplied from the PS boundary (left boundary). The head particle and the tail particles of an agent are indicated by blue and magenta colors, respectively.

https://elifesciences.org/articles/84749/figures#video7

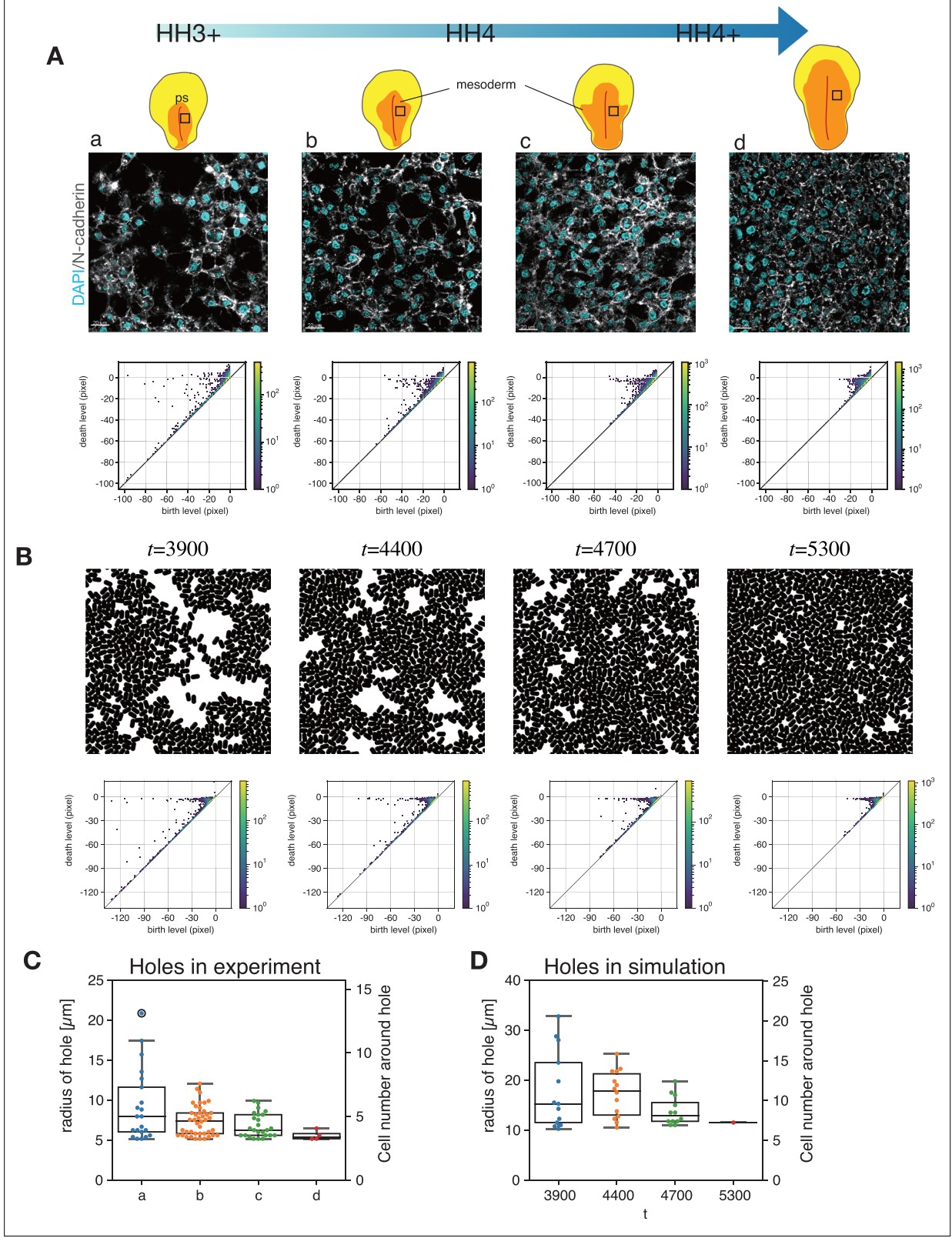

**Figure 6.** Changes in the meshwork structure during development. (**A**) Spatial distribution of the cells in the mesoderm tissue stained for nuclei (cyan) and N-cadherin (white) at different developmental stages (top) and the corresponding persistence diagrams (PD) (bottom). The persistent homology analysis was performed using binary images. The pixel size is 0.215 μm. (**B**) Simulation with the supply of the agents where the supply rate increases with time. Snapshots (top) and the corresponding PD. (**C**) The radius of holes that appear in the hole branch in the PD in (**A**). (**D**) The radius of holes that

*Figure 6 continued*

appear in the hole branch in the PD in (**B**). Holes are identified as the points with a birth level smaller than –5 μm (**A**) and –10 μm (**B**) and a death level larger than –2.5 μm (see Method). The color in the PD diagram (AB) indicates the multiplicity of the points.

The online version of this article includes the following source data for figure 6:

**Source data 1.** Numerical data plotted in *Figure 6C*.

**Source data 2.** Numerical data plotted in *Figure 6D*.

that the size of the holes was initially large, but gradually decreased with time (*Figure 6B*). The radius of holes obtained from the PDs (*Figure 6B* bottom) decreases as well (*Figure 6D*), although the hole sizes and the number of cells surrounding the holes (*Figure 6C and D*, left y-axis) in the simulation were slightly larger than those of the experiment. This quantitative difference might come from the fact that the shape of the cells in the simulation is kept constant, while the shape of the real mesoderm cells changes dynamically. From these results, one possible reason for the decrease in the hole size as the developmental stage of the embryo proceeds could be a potential increase in the appearance rate of mesoderm cells at the primitive streak.

## Discussion

Our results revealed a novel mode of collective cell migration, in which the migrating nascent mesoderm cells form a dynamic meshwork structure in three-dimensional space between the epiblast and endoderm while moving collectively in the anterior-lateral direction. In the early gastrulation stage of chick embryos, the fate of the various cell populations in the mesoderm has been studied in detail and it is known to be determined by the final migration destination (*Psychoyos and Stern, 1996*). However, it was not well understood whether the mesoderm cells move collectively without scattering toward their destination. In addition, the mesoderm cells were thought to be densely packed without any spatial structure (*Psychoyos and Stern, 1996*; *Yang et al., 2002*). In this study, we investigated these points quantitatively by the 3D time lapse imaging and the horizontal thin optical sectioning of the mesoderm region in fixed whole mount embryos applying tissue-clearing method. From the analysis of the multicellular tracking data, we confirmed that the cells emerging from the primitive streak migrate collectively with the characteristic decay length of about 60 μm. The cell trajectories we analyzed are presumed to be almost entirely mesodermal, although a small fraction of definitive-endoderm cells may be included near the rostral part of the primitive streak (*Psychoyos and Stern, 1996*). From the horizontal thin sections, we found that the cells in the mesoderm region form a meshwork structure. The diameter of the holes is about 30 μm, which is almost comparable to the characteristic decay length of the collective migration. From these results, we presume that this meshwork structure is relevant to the collective migration of the cells in the mesoderm region. Since only little extracellular matrix exists in the mesoderm (*Van Hoof et al., 1986*; *Sanders, 1986*; *Sanders, 1979*), the formation of meshwork structure should be based on the intercellular adhesion. In fact, the disruption of the extracellular domain using a mutant form of N-cadherin lead the exclusion tendency of the mutant cells from the meshwork of the control cells. Moreover, although the migration speed along trajectory is unaltered, the directionality of individual cell migration, the tissue progression speed, and the stability of the direction of collective motion were reduced for the mutant cells compared to the control cells. These results suggest that the cell-cell adhesion coordinates the migration of the mesoderm cells. To summarize, we conclude that the cell-cell adhesion plays a fundamental role in the meshwork formation for the mesoderm cells to migrate collectively. Such collective motion could contribute to the robust formation of cell migration pattern in response to guidance signals such as chemoattractant and chemorepellent (*Yang et al., 2002*). Since the introduced deletion mutant of N-cadherin lacks the extracellular domain that is responsible for cell-cell adhesion activity, it is expected that the cells expressing the mutant N-cadherin have reduced cell-cell adhesion. However, the mutant N-cadherin may also affect other processes such as cell motility and signaling, which requires further investigations.

Extracting information about the organization and arrangement of cells in tissues from microscopy images and comparing them with mutants has been largely based on visual inspection. Moreover, their quantitative and objective characterization is often challenging because of their variability and

lack of periodicity. To obtain the information of the holes of the meshwork structure in the mesoderm tissues, such as their size and position objectively and automatically, we used persistent homology, a tool of topological data analysis (TDA). TDA is a recently growing unique methodology, and it provides geometric information of the complex data, which has been employed in physical, medical, and biological research (*Hiraoka et al., 2016*; *Lawson et al., 2019*; *McGuirl et al., 2020*; *Topaz et al., 2015*). We used this method to extract the information of the dynamics of the meshwork structure, which is still a challenging task in TDA. By applying the same analysis method to the simulation result, we compared it with the experiment quantitatively and we successfully showed that the theoretical model captures the essential aspect of the meshwork formation observed experimentally.

The in vivo collective migrations of mesenchymal cells have been reported for the neural crest cells in frog and chick (*Li et al., 2019*; *Szabó and Mayor, 2018*). These neural crest cells migrate on a two-dimensional surface within a confined space with a physical barrier of neighboring tissues (*Szabó and Mayor, 2018*). Contact attraction (*Hayakawa et al., 2020*; *Li et al., 2019*) and contact inhibition (*Carmona-Fontaine et al., 2008*) orient the cell motion to induce the collective cell migration, such as chain migration and stream formation (*Szabó et al., 2019*). In contrast, mesoderm cells at the early gastrulation stage migrate in the three-dimensional space between the epiblast and endoderm without physical barrier in the lateral direction. Almost all cells were attached to other cells. Upon collision, the mesoderm cells stay in contact for more than a few tens of minutes (*Figure 4—figure supplement 3*). Thus, contrary to the neural crest cells, the mesoderm cells did not show contact inhibition of locomotion, which is consistent with the case of the mouse mesoderm cells (*Saykali et al., 2019*). In situations where cells exploit other cells as scaffolds and the cell density is low, we speculate that forming a meshwork rather than a three-dimensional mass would be more efficient to extend the distance.

In the nascent mesoderm tissue of chick embryo, matured ECM is almost absent in the intermediate layer where cells are in contact with other cells but not with the epiblast or endoderm (*Van Hoof et al., 1986*; *Sanders, 1986*; *Sanders, 1979*). How cells in the intermediate layer can generate traction force for the movement is an intriguing future question. Mesoderm cells on the basal lamina of either epiblast or endoderm can generate traction force to migrate. By adhering to these cells, it may be possible that the mesoderm cells in the intermediate layer move forward together in a passive manner. In addition to such passive movement, the intermediate cells might generate active force at the intercellular contacts, by which they migrate further. Another possibility for the active process of the cell motility is the treadmilling of intercellular junction, which has been implicated in the migration of adhering cells (*Peglion et al., 2014*).

During the vasculogenesis, endothelial cells also form a meshwork structure with cords of cells that surround the regions void of cells. In this case, cell aggregates formed initially are connected to organize into a primitive vascular plexus (*Risau and Flamme, 1995*). The cell motions appear to be random along the cords (*Sato et al., 2010*). In contrast, the meshwork structure we observed is formed by the mesoderm cells which are provided from the primitive streak without the formation of cords of cells and a different lineage than the cells that contribute to the vasculogenesis. Moreover, the mesoderm cell motion is biased to the anterior-lateral direction. Thus, although there are some similarities in the 2D horizontal section patterns, the 3D structures are different between these two cases.

During the development of enteric nervous system, enteric neural crest cells (ENCCs) migrating in the mesenchyme also form a meshwork structure within a narrow 2-dimensional layer (*Newgreen et al., 2013*). ENCCs migrate in chains and the cells immediately behind the preceding chains often follow the same path (*Young et al., 2004*). Thus, the network created by the preceding cells often remained intact for many hours. This constant shape of the network contrasts with the dynamic properties of the meshwork structure formed by the mesodermal cells in the chick embryo.

To understand how the meshwork forms, we developed a theoretical model that demonstrated that the elongated shape of agents and the attractive interaction between them are the key factors for the formation of meshwork. While a previous study reported that the branches of a meshwork structure showed nematic order (*Palm and Merks, 2013*), it was not clear how a meshwork structure emerges as the density increases. We showed quantitatively that clusters composed of agents deforms in the direction of the nematic order of the agent elongation. As the density increases, the elongated clusters grow and finally fuse with each other to form a meshwork structure. Although the agents in the current model do not deform their shape, actual mesoderm cells do, which enables them

to migrate. Presumably, the shape deformation may play an important role for the 3D meshwork structure formation where the intermediate cells have no scaffold to migrate other than other cells, like the one that we found in the mesoderm of chick embryo. It is thus a future work to investigate how the shape deformation of the agents contributes to the 3D meshwork structure formation.

Our data and theoretical model suggest that cell elongation and cell-cell adhesion actively stabilize the meshwork. There may still be a possibility that a spinodal-like instability driven by N-cadherin-mediated adhesion. If it exists, anisotropic cell shape would act on top of the instability process. Future work will be required to determine whether such an instability exists in the meshwork formation of chick mesoderm.

# Materials and methods

**Key resources table**

| Reagent type (species) or resource | Designation | Source or reference | Identifiers | Additional information |
|---|---|---|---|---|
| Biological sample (Chicken) | Wild Type Chicken Eggs | Shimojima farm, Kanagawa, JP | | |
| Biological sample (Chicken) | Tg (pLSi/ΔAeGFP) Chicken Eggs | ABRC, University of Nagoya; *Motono et al., 2010* | | |
| Antibody | anti-E-Cadherin (mouse monoclonal) | BD Transduction Lab | Cat# 610181; RRID:AB_397580 | IF(1:1000) |
| Antibody | Anti-N-cadherin, (rabbit polyclonal) | Takara bio | Cat# M142; RRID:AB_444317 | IF(1:300) |
| Antibody | anti-N-Cadherin/A-41CAM Clone GC-4 (mouse monoclonal) | Sigma-Aldrich | Cat# C2542; RRID:AB_258801 | IF(1:50) |
| Antibody | anti-Mouse IgG, (H+L) Highly Cross-adsorbed Antibody, Alexa Fluor 488 (Goat) | Thermo Fisher | Cat# A-11029; RRID:AB_138404 | IF(1:300) |
| Antibody | anti-Rabbit IgG, (H+L) Highly Cross-Adsorbed Secondary Antibody, Alexa Fluor 488 (Goat) | Thermo Fisher | Cat# A-11034; RRID:AB_2576217 | IF(1:300) |
| Chemical compound, drug | Cellstain DAPI Solution | Dojindo Laboratories | Cat# D523 | 1:100 dilution |
| Recombinant DNA reagent | pGEM-T Easy | Promega | Cat# A1360 | |
| Recombinant DNA reagent | pCAG-H2B-eGFP | Dr. Hadjantonakis (MSKCC, NY). | | |
| Recombinant DNA reagent | pCAG-N-Cad-M-2A-H2B-mCherry | This study | | See Materials and methods 'Generation of chick-N-Cadherin mutant' |
| Recombinant DNA reagent | pCAG-N-Cad-M-2A-eGFP-CAAX-2A-H2B-mCherry | This study | | See Materials and methods 'Generation of chick-N-Cadherin mutant' |
| Sequence-based reagent | N-cadherin cloning primer Fw 5'-ATGTGCCGGATAGCGGGAAC-3' | Hokkaido System Science Co., Ltd | | |
| Sequence-based reagent | N-cadherin cloning primer Rev 5'-TCAGTCATCACCTCCACCG-3' | Hokkaido System Science Co., Ltd | | |
| Sequence-based reagent | N-Cad-M primer 1 Fw 5'- ATGGGTTCTTCTAAATCTAAACCAAAAGAT CCATCTCAACGTATGAAGCGCCGTGATAAGG-3' | Fasmac | | |

*Continued on next page*

*Continued*

| Reagent type (species) or resource | Designation | Source or reference | Identifiers | Additional information |
| --- | --- | --- | --- | --- |
| Sequence-based reagent | N-Cad-M primer 1 Rev 5'-GTCATCACCTCCACCGTAC-3' | Fasmac | | |
| Sequence-based reagent | N-Cad-M primer 2 Fw 5'-GCGGCCGCGGATCCGCATGCGCCACCATGGGTTCTTCT-3' | Thermo Fisher | | |
| Sequence-based reagent | N-Cad-M primer 2 Rev 5'-TTGCTCACCATAACGCATGCTTTAGGTCCAGGGTTCTCC-3' | Thermo Fisher | | |
| Software, algorithm | HomCloud | *Obayashi et al., 2018* | Ver. 3.0.1 | |
| Software, algorithm | IMARIS | Oxford instruments, UK | Ver 9.5.1 | |
| Software, algorithm | Matlab | Mathworks Inc, Natick, MA | R2024b | |

## Chick embryo collection and ex vivo culture

Fertilized hen's egg (Shimojima farm, Kanagawa, Japan) or fertilized transgenic chicken's eggs (Avian Bioscience Research Center at Nagoya University) (Key resources table) were incubated at 38.5 °C until embryos reached the desired developmental Hamburger-Hamilton stage (*Hamburger and Hamilton, 1951*).

## Electroporation

For electroporation, expression vectors were injected between the epiblast and vitelline membrane of embryos at a concentration of 2–5 µg/ul and electroporated with 1 mm platinum electrodes by using an electroporator (NEPA21 Super Electroporator; Nepagene) with the following parameters: 8.0 V, 0.5 ms width, one poring pulse, followed by 5.0 V, 25.0 ms width, 50 ms interval, five polarity exchanged transfer pulses. Embryos were then cultured for several hours according to the Easy Culture (EC) protocol (*Chapman et al., 2001*).

## Generation of chick-N-Cadherin mutant

Full-length of N-cadherin coding sequence (accession number NM_001001615.1) was amplified by PCR from cDNA of HH 5–7 chick embryos using the following primers: Fw 5'-ATGTGCCGGATAGCGGGAAC-3' and Rev 5'-TCAGTCATCACCTCCACCG-3', which was subcloned into the pGEM-T Easy vector (Promega). The full length of N-cadherin fragment was then used as a template to generate an N-cadherin mutant lacking the extracellular and transmembrane domains, which corresponds to amino acids 752–912 of the N-cadherin protein. To ensure membrane localization of N-cadherin mutants, the second PCR reaction was performed using the following primers in which the sequence of an N-myristoylation signal from Src kinase (*Kaplan et al., 1988*) was added at the 5' side of the forward primer: Fw 5'-ATGGGTTCTTCTAAATCTAAACCAAAAGATCCATCTCAACGTATGAAGCGCCGTGATAAGG-3', and Rev 5'-GTCATCACCTCCACCGTAC-3'. This amplified fragment was named N-Cad-M and was subcloned into the 5' side from P2A peptide (ATNFSLLKQAGDVEENPGP) of the pCAG-P2A-H2B-mCherry vector by In-Fusion Cloning (Takara, Japan). To visualize the membrane of cells that express N-Cad-M, the N-Cad-M-P2A was amplified by PCR with a DNA fragment set of 5'-GCGGCCGCGGATCCGCATGCGCCACCATGGGTTCTTCT-3' and 5'- TTGCTCACCATAACGCATGCTTTAGGTCCAGGGTTCTCC-3', which was then subcloned into the 5' side from the eGFP sequence of the pCAG-eGFP-CAAX-P2A-H2B-mCherry expression vector by In-Fusion Cloning. The above oligonucleotides used in this study are listed in key resources table and the recombinant DNA constructed in this study are summarized in key resources table.

## Immunohistochemistry

For immunohistochemistry, embryos are fixed in 4% PFA, and the following antibodies were used: Purified Mouse Anti-E-Cadherin (610181, BD Transduction Lab); Anti-N-Cadherin, polyclonal (Code

No. M142, Takara Bio); Monoclonal Anti-N-Cadherin/A-CAM (Clone GC-4, Product No. C2542, Sigma-Aldrich). Alexa Fluor secondary antibody (Goat anti-Rabbit IgG, Alexa Fluor 488, A-11034; Goat anti-Mouse IgG, Alexa Fluor 488, A-11029, Thermo Fisher Scientific) were used for double color detection. DAPI (Cellstain DAPI Solution, 1:100, 340–07901 Dojindo Laboratories) for the labeling the nucleus was used. After washing, embryos were cleared with SeeDB-2G solution (*Ke et al., 2016*) before being processed for imaging. Immunofluorescence images were captured with a laser scanning confocal microscope (FV3000RS with IX83 inverted; Olympus) equipped with UPLSAPO 30xS/1.05 NA, 60xS/1.3 NA objective lenses, using Fluoview (Olympus) as the image acquisition software. For each embryo, several images corresponding to different focal planes and different fields were captured using z-section and tilling functions. The acquired images were imported to Imaris 9.5.1 (Oxford Instruments, UK) to 3D-visualize for further analysis. The antibodies used in this study are summarized in key resources table.

## in vivo live imaging

For in vivo live imaging, the H2B-eGFP-expressing WT chick embryos or the transgenic-GFP chick embryos was transferred dorsal side up on glass-base dish (Iwaki, 3910–035) with semi-solid albumin/ agarose (0.1%). Embryos were imaged at 38.5 °C using an inverted multi-photon microscopy (Olympus MP, FVRS-F2SJ) coupled to a Maitai DeepSee HP laser at 890 nm wavelength and an InSight DeepSee laser at 1100 nm using 25 x/water 1.05 NA long distance objective lens (XLPLN25XW-MP).

## Obtaining the trajectory of individual mesoderm cells

To obtain the trajectories of mesoderm cells, live imaging data of embryo expressing H2B-eGFP were analyzed using IMARIS (Oxford Instruments). The movement of each nucleus was identified using '*Spot*' function in the package 'IMARIS for tacking' as described below. For identifying the nuclear position, we used '*Spot Detection*' with the parameter '*Estimated Diameter*' to be 6 μm by adjusting the lowest threshold in '*Quality*' setting in the '*Filter*' section to a value with which the faintest nuclei were reliably distinguished from the background. For tracking, '*Autoregressive Motion*' were used in the '*Algorithm*' section with '*Max Distance*' to be 8 μm and '*Max Gap Size*' to be 3 without using '*Fill gaps with all detected objects.*' We then removed the short tracks by applying the '*Track Duration above 1800 s*' in the '*Classify Tracks*' section. The trajectory data obtained in the above way was then exported as a comma-separated values (csv) file for the further analysis. Then, the mean square displacement, directionality, polar order parameter, and mean square relative distance as described in the following sections were obtained using a custom-made code of Matlab (Mathworks Inc, Natick, MA).

## Individual cell speed

The instantaneous velocity of each cell is defined by the displacement of the cell position in two subsequent images divided by the time interval. The individual cell velocity is calculated by averaging the instantaneous velocity over the trajectory. The individual cell speed is the magnitude of the individual cell velocity.

## Progression velocity and progression speed

To calculate the tissue progression speed, the image window is divided into small regions of 50 μm × 50 μm as in *Figure 1D*. Then, the progression velocity is calculated as the temporal average of the average velocity of cells in each region at each time point. The progression speed is the magnitude of the progression velocity.

## Directionality

The directionality was calculated using the formula given by

$$\text{Directionality} = \langle d/D \rangle$$

where $d$ is the start-to-end distance and $D$ is the actual length of trajectory between the start point and the end point. The bracket $\langle \cdot \rangle$ indicates the average over the trajectories of the cells in a sample. The value of directionality depends on the time interval of the trajectory. In this paper, we consider

the trajectories for 20 min. The directionality is close to one when the motion is in a straight trajectory, while it is close to zero when the motion is random or when the trajectory forms a closed loop.

## Mean squared displacement

For each sample, the mean squared displacement (MSD) was calculated for individual migrating cells and then average them over ensemble. The MSD for a given sample was calculated using the formula, given by

$$MSD\,(t) = \frac{1}{N\,(T-t)} \sum_{i=1}^{N} \sum_{\tau=1}^{T-t} \left\{ \mathbf{r}_i\,(\tau+t) - \mathbf{r}_i\,(\tau) \right\}^2,$$

where $\mathbf{r}_i\,(\tau)$, $t$, $T$, and $N$ are the 3D position of cell $i$ at time $\tau$, the lag time, final time, and number of trajectories in the sample, respectively. To obtain the exponent $\alpha$ of MSD, we fitted MSD(t) with the curve $Dt^{\alpha}$ where $D$ is a coefficient. For the fitting, we used lsqcurvefit of Matlab R2021b (MathWorks). The exponent $\alpha$ is 1 for random motion, while it is close to 2 if the motion is ballistic (straight). For $1 < \alpha < 2$, the motion is known as super-diffusion.

## Auto-correlation function of velocity

For each sample, the auto-correlation function (ACF) of velocity was calculated for individual migrating cells and then average them over ensemble. The ACF of velocity for a given sample was calculated using the formula, given by

$$\text{ACF}\,(t) = \frac{1}{N\,(T-t)} \sum_{i=1}^{N} \sum_{\tau=1}^{T-t} \mathbf{v}_i\,(t+\tau) \cdot \mathbf{v}_i\,(\tau) \Big/ \frac{1}{NT} \sum_{i=1}^{N} \sum_{\tau=1}^{T} \mathbf{v}_i\,(\tau) \cdot \mathbf{v}_i\,(\tau)$$

where $\mathbf{v}_i\,(\tau)$, $t$, $T$, and $N$ are the velocity vector of cell $i$ at time $\tau$, the lag time, final time, and number of trajectories in the sample, respectively. The ACF of velocity approaches to zero for sufficiently long time if there is no bias in the migration direction.

## Polar order parameter

For the trajectories obtained by the tracking analysis, the polar order parameter at a given time was calculated using the formula, given by

$$\varphi\,(t) = \left| \frac{1}{N} \sum_{i=1}^{N} \frac{\mathbf{v}_i\,(t)}{|\mathbf{v}_i\,(t)|} \right|,$$

where $N$ is the number of tracked cells, and $\mathbf{v}_i\,(t)$ is the instantaneous cell velocity of cell $i$. The polar order parameter $\varphi\,(t)$ is close to one if all cells move in the same direction, while it is close to zero if cells move in a random direction. For the data shown in *Figure 1J*, we calculated the temporal average of $\varphi\,(t)$ in entire region (500 μm × 500 μm × z-depth). For the data in *Figure 4J*, *Figure 4—figure supplement 2D*, since the size of imaged region was different between embryo samples, we divided the imaged region into subareas (125 μm × 125 μm × z-depth), in each of which we measured the temporal average of $\varphi\,(t)$. Then, they were averaged in each embryo.

## Mean squared relative distance (MSRD)

We took a pair of cells which were initially at the distance less than 20 μm, supposing that these cells were in contact with each other at that moment. Then, the mean squared relative distance (MSRD) was calculated for the pairs using the following formula,

$$\text{MSRD}\,(t) = \frac{1}{N\,(N-1)\,/2} \sum_{j=i+1}^{N} \sum_{i=1}^{N} \left| (\mathbf{r}_i\,(t) - \mathbf{r}_j\,(t)) - (\mathbf{r}_i\,(0) - \mathbf{r}_j\,(0)) \right|^2$$

where $\mathbf{r}_i\,(t)$, and $N$ are the 3D position of cell $i$ at time $t$, and number of trajectories in the sample, respectively.

## Topological structure analysis using persistent homology

To characterize the meshwork structure in the mesoderm quantitatively, we focused on the holes void of cells. To this end, we performed persistent homology analysis by using the software named HomCloud (3.0.1) (*Obayashi et al., 2018*). We first prepared a black and white binary pixel image from the original image by thresholding, where the pixels occupied by cells are white. The binary image was then used as an input data for HomCloud. Each topological structure is characterized by a pair of two values called birth and death levels based on the Manhattan distance from the interface between white and black pixels. Thus, birth and death levels are given in the unit of pixels. These two quantities basically represent the size of the identified topological structures and the distance between two topological structures, respectively. The birth and death levels are usually called birth and death times, respectively, in the persistent homology field. In this paper, to avoid confusion we use the term 'level' instead of 'time.' In HomCloud, the identified topological structures are visualized in persistence diagram (PD), which plots each pair of birth and death levels. Since holes are identified by the 0th persistent homology, we focused on 0th PDs, i.e., the PDs for the 0th persistent homology. Each birth-death pair characterizes a black region in the binary image (*Figure 2E*; *Obayashi et al., 2018*). The difference between the death and birth levels is called persistence, which is often referred to as lifetime in the field of persistent homology. The topological structures with small persistence are basically noise (*Obayashi et al., 2018*). Although in the original PD the points correspond to actual holes appear in the region with birth level <0 and death level >0, the death level of most holes in the experimental image becomes slightly smaller than 0 due to fluctuations possibly caused by several factors, including those in staining and fluorescent imaging. By taking this into consideration, we identified the points with a birth level smaller than –10 μm and a death level larger than –2.5 μm as detected holes, except for the those in *Figure 6C* where the threshold is set as the birth level smaller than –5 μm and the death level larger than –2.5 μm because there was no hole satisfying the above stricter threshold for the later stage (*Figure 6Ac* and 6Ad). Since the magnitude of birth level corresponds to the shortest distance from the center to the periphery of a hole, we regarded this multiplied by the length of a pixel as the radius of the hole. The number of cells that surround each single hole was calculated from the perimeter length by assuming that a cell diameter is 10 μm.

## Analyzing the dynamics of holes

To visualize the spatiotemporal dynamic of holes, we first carried out the inverse analysis by HomCloud (*Obayashi et al., 2018*) for a total of 13 images at 2 min intervals out of the live imaging of 24 min and saved them as a series of images (*Figure 3—figure supplement 1D* bottom). From this 2D image sequence, we constructed a z-stack image using the 3D image reconstruction function of IMARIS by setting the z-interval at 5 μm. Each hole was visualized by the '*surface*' function in IMARIS. To ensure each hole was visualized individually and the adjacent holes were reliably split, we used the following parameters. We set '*Threshold*' to 130, enabling '*Split touching Objects (Region Growing)*' and the value of the '*Estimated Diameter*' to 10 μm. We used '*Classify Seed Point*' for the filter type in '*Quality*' section with '*Lower Threshold* set at 40. We manually chose five representative holes during 24 min of observation as shown in *Figure 3D*.

## Autocorrelation function of the direction of collective migration

The direction of collective migration $P(t)$ is defined from

$$\varphi(t)\,\hat{\mathbf{P}}(t) = \frac{1}{N}\sum_{i=1}^{N}\frac{\mathbf{v}_i(t)}{|\mathbf{v}_i(t)|},$$

where $\varphi(t)$ is the polar order parameter, $N$ is the number of tracked cells, and $\mathbf{v}_i(t)$ is the instantaneous cell velocity of cell $i$. The autocorrelation function of the direction of collective migration $\hat{\mathbf{P}}(t)$ is given by

$$\mathrm{ACF}(t) = \left\langle \hat{\mathbf{P}}(t) \cdot \hat{\mathbf{P}}(0) \right\rangle$$

where $\langle \cdot \rangle$ indicates the average over ensemble. The direction of collective migration $\hat{\mathbf{P}}(t)$ and its auto-correlation function $\mathrm{ACF}(t)$ were calculated for the cells in small regions of 125µm x 125µm along x- and y-coordinates, which is averaged for each sample.

## Measurement of aspect ratio

To obtain the aspect ratio of cell shape (*Figure 4—figure supplement 1*), we rendered fluorescently labeled cell membrane using the 'surface' function in IMARIS. The shortest length and the longest length were obtained from object-oriented Bounding Box OO statistical variables of IMARIS (*Figure 4—figure supplement 1*, top right). The aspect ratio was then calculated by dividing the longest length by the shortest length (*Figure 4—figure supplement 1*, bottom right).

## Theoretical Model for the formation of meshwork-like structure

In order to understand how the mesoderm cells organize into the meshwork structure, we introduce a mathematical model where each cell is represented by a self-propelled rod-shaped agent. To take into account the adhesion and volume exclusion between the cells, a short-range attractive interaction with a repulsive core is assumed between the agents. Since the typical size and migration speed of the cells are about 10 *µm* and 3 *µm/min*, we can assume that their dynamics is in the overdamped regime. Then, the equation of motion of the agent $i$ is given by

$$\gamma \frac{d\mathbf{r}_{i,p}}{dt} = \mathbf{F}_{i,p}^{\mathrm{str}} + \mathbf{F}_{i,p}^{\mathrm{act}} + \mathbf{F}_{i,p}^{\mathrm{cell-cell}} + \boldsymbol{\xi}_{i,p} \tag{1}$$

Here, the actual degrees of freedom for each rod-shaped agent are given by the head ($p = 2$) and tail ($p = 1$) particles of the diameter $d$ that are separated by the length $l_c$, which gives the aspect ratio of the agent shape as $r = (l_c + d)/d$. $\mathbf{r}_{i,p}$ is the position of the tail and head particles of the agent $i$, and the friction coefficient $\gamma = 3\pi\eta(3d + 2l_c)/5$ takes into account the effect of the elongated shape with the effective viscosity $\eta$. The agent shape is kept the same by the stretching elasticity acting between the head and tail particles:

$$\mathbf{F}_{i,p}^{\mathrm{str}} = \frac{\kappa^{\mathrm{str}}}{l_c}\left(|\mathbf{r}_{i,2} - \mathbf{r}_{i,1}| - l_c\right)\frac{\mathbf{r}_{i,2} - \mathbf{r}_{i,1}}{|\mathbf{r}_{i,2} - \mathbf{r}_{i,1}|}\left(\delta_{p,1} - \delta_{p,2}\right). \tag{2}$$

Here, $\delta_{pq}$ is the Kronecker delta that takes 1 if $p = q$ and 0 otherwise. A constant effective self-propulsion force of the magnitude $f^{\mathrm{act}}$ is assumed to act only on the head particle as,

$$\mathbf{F}_{i,p}^{\mathrm{act}} = f^{\mathrm{act}}\frac{\mathbf{r}_{i,2} - \mathbf{r}_{i,1}}{|\mathbf{r}_{i,2} - \mathbf{r}_{i,1}|}\delta_{p2}. \tag{3}$$

To implement the interaction between the agents, each rod-shaped agent is discretized into $M$ helper particles, including the head and tail particles, of the equal distance less than $\frac{3}{6}\sigma^{\mathrm{cc}}$, and the interaction force is imposed between the closest helper particles of a pair of agents (see below). The force on the th helper particle is imposed on the head and tail particles with the geometric weight $1 - \alpha_p$ and $\alpha_p$, where $\alpha_p|\mathbf{r}_{i,2} - \mathbf{r}_{i,1}|$ is the distance of the th particle and the tail particle. As a result, the interaction force on particle of agent is given by

$$\mathbf{F}_{i,p}^{\mathrm{cell-cell}} = \sum_j\left(\sum_{p'=1}^{M}\mathbf{f}_{i,p':j,q}^{i,p} - \sum_{q=1}^{M}\mathbf{f}_{j,q:i,p'}^{i,p}\right). \tag{4}$$

Here, $\mathbf{f}_{i,p':j,q}^{i,p} = \left((1 - \alpha_{p'})\delta_{p1} + \alpha_{p'}\delta_{p2}\right)\mathbf{F}^{\mathrm{cube}}\left(\mathbf{r}_{i,p'} - \mathbf{r}_{j,q}; \sigma^{\mathrm{cc}}, \xi^{\mathrm{cc}}, \epsilon_{\mathrm{rep}}, \epsilon_{\mathrm{atr}}\right)$, where $q$ is the particle index of agent $j$ that is the closest to particle $p'$ of agent $i$. Here, we use the following function of the short-range attraction with the repulsive core with the cutoff distance $r < \sigma^{\mathrm{cc}} + \xi^{\mathrm{cc}}$ (*Schnyder et al., 2020*):

$$F^{\text{cube}}\left(\mathbf{r}; \sigma^{\text{cc}}, \xi^{\text{cc}}, \epsilon_{\text{rep}}, \epsilon_{\text{atr}}\right) = \begin{cases} 0 & \left(r - \sigma^{\text{cc}} \leq -\xi^{\text{cc}}\right) \\ \left(\epsilon_{\text{rep}} + \epsilon_{\text{atr}}\right) g'\left(-\left(r - \sigma^{\text{cc}}\right)\right)\dfrac{\mathbf{r}}{r} & \left(-\xi^{\text{cc}} \leq r - \sigma^{\text{cc}} \leq 0\right) \\ -\epsilon_{\text{atr}} g'\left(r - \sigma^{\text{cc}}\right)\dfrac{\mathbf{r}}{r} & \left(0 \leq r - \sigma^{\text{cc}} \leq \xi^{\text{cc}}\right) \\ 0 & \left(\xi^{\text{cc}} \leq r - \sigma^{\text{cc}}\right) \end{cases} \tag{5}$$

where $g'\left(r\right) = \frac{6r}{\xi^3}\left(\xi - r\right)$ is the derivative of $g\left(r\right) = \frac{r^2}{\xi^3}\left(3\xi - 2r\right)$. Finally, $\boldsymbol{\xi}_{i,k}$ is a Gaussian white noise with zero mean and $\langle \xi_{i,k,\alpha} \xi_{j,l,\beta} \rangle = \sigma \delta_{ij}\delta_{kl}\delta_{\alpha\beta}$ with the noise strength $\sigma$. The schematics of the model and the potential and the force profiles of the attractive interaction are shown in *Figure 5—figure supplement 1*.

To understand the essential aspect of the meshwork formation, we consider the model in a two-dimensional space in the range $-L_x/2 \leq x \leq L_x/2$, and $-L_y/2 \leq y \leq L_y/2$, where $L_x$ and $L_y$ are the system size. For the steady-state analysis, the periodic boundary conditions are assumed in both $x$ and $y$ directions. In the case where the cells are supplied from one $x$ boundary in the manner as described below, the periodic boundary condition is assumed only in the $y$ directions.

To mimic the experimental situation where the cells are supplied from the primitive streak, we prepared the source of agents at $x = -L_x/2$ from which the agents are supplied at random $y$ position at constant rate $r_{\text{source}}$. In the source, the agents undergo random walk, without self-propulsion nor interaction with other agents, in a harmonic potential centered at $x_{\text{source}} = -\left(L_x + L_{\text{source}}\right)/2$ that keeps the agents in the source. Here, $L_{\text{source}} = 2l_c$ is the width of the source. The agents that are supplied from the source experience the repulsive interaction from the source within the cutoff distance $\frac{1}{2}\left(x_{i,1} + x_{i,2}\right) < x_{\text{source}} + L_{\text{source}}$,

$$\mathbf{F}^{\text{source}}_{i,p} = \begin{cases} k^{\text{source}}\left(x_{\text{source}} + L_{\text{source}} - \frac{1}{2}\left(x_{i,1} + x_{i,2}\right)\right), & \left(\frac{1}{2}\left(x_{i,1} + x_{i,2}\right) < x_{\text{source}} + L_{\text{source}}\right) \\ 0, & \left(\frac{1}{2}\left(x_{i,1} + x_{i,2}\right) \geq x_{\text{source}} + L_{\text{source}}\right) \end{cases}, \tag{6}$$

in addition to the self-propulsion and interaction force. Furthermore, in this case, we introduce the chemotactic force.

$$\mathbf{F}^{\text{chemotaxis}}_{i,p} = \mathbf{f}^{\text{chemotaxis}}\left(1 - \hat{\mathbf{x}} \cdot \frac{\mathbf{r}_{i,2} - \mathbf{r}_{i,1}}{\left|\mathbf{r}_{i,2} - \mathbf{r}_{i,1}\right|}\right)\left(\delta_{p2} - \delta_{p1}\right)\hat{\mathbf{x}} \tag{7}$$

which rotates the agents so that they tend to move away from the source. The other $x$ boundary is the sink of agents. That is, when the agents reach the boundary at $x = L_x/2$, the agents are taken away

**Table 1.** The parameters used in the numerical simulation.

| Variables | Symbols | Values |
|---|---|---|
| Cell diameter | $d$ | 1 |
| Cell head and tail length | $l_c$ | $d\left(r - 1\right)$ |
| Cell stretching elasticity | $\kappa^{str}$ | 10 |
| Cell-cell attraction strength | $\epsilon^{atr}$ | 0.001 |
| Cell-cell repulsion strength | $\epsilon^{rep}$ | 0.1 |
| Cell-cell interaction length | $\sigma^{cc}$ | $d$ |
| Width of cell-cell attraction well | $\xi^{cc}$ | $d/2$ |
| Effective viscosity | $\eta$ | 0.1 |
| Temperature | $k_B T$ | 0.004142 |
| Noise strength | $\sigma$ | $2k_B T\gamma$ |
| Number of cells | $N_{cell}$ | 1600 |

**Table 2.** The parameters used in the numerical simulation with the agent supply.

| Variables | Symbols | Values |
|---|---|---|
| Self-propulsion force | $f^{act}$ | 0.01 |
| Repulsive strength from source | $k^{source}$ | 0.1 |
| Chemotactic force | $f^{chemotaxis}$ | 0.004 |
| Simulation box size in x | $L_x$ | 60 |
| Simulation box size in y | $L_y$ | 40 |

from the system and placed back to the source. Therefore, the equation of motion of the agent $i$ in this case is given by

$$\gamma \frac{d\mathbf{r}_{i,p}}{dt} = \mathbf{F}_{i,p}^{str} + \mathbf{F}_{i,p}^{act} + \mathbf{F}_{i,p}^{cell-cell} + \mathbf{F}_{i,p}^{source} + \mathbf{F}_{i,p}^{chemotaxis} + \boldsymbol{\xi}_{i,p} \qquad (8)$$

The parameters that were used in the numerical simulations are summarized in *Table 1*. In the case that the cells are supplied from the source, additional parameters are summarized in *Table 2*. The equation of motion was solved using the Euler method.

In the analysis shown in *Figure 5F*, to identify clusters of agents, we applied the Cluster analysis modifiers of Ovito Pro (*Stukowski, 2010*) to the simulation data, including all the helper particles with the cutoff length $\sigma^{cc} + \xi^{cc}/2$. To quantify the elongation of the cluster, we measured the gyration tensor of each cluster defined by

$$S_{\alpha\beta} = \langle \tilde{r}_{i,p,\alpha} \tilde{r}_{i,p,\beta} \rangle_{i,p} , \qquad (9)$$

where $\tilde{r}_{i,p,\alpha}$ is the $\alpha$ component of the position of helper particle $p$ of agent $i$ measured from the center of the cluster, the average $\langle \cdot \rangle_{i,p}$ is calculated over helper particles $p$ of all agents $i$ that belong to the cluster. By using this gyration tensor, we calculated the cluster aspect ratio and the longitudinal angle as the square root of the ratio of the two eigenvalues, $\sqrt{\lambda_+/\lambda_-}$ ($\lambda_+ \geq \lambda_-$), and as the direction of the major principal axis, respectively. To eliminate small clusters, we took into account only the clusters composed of more than four cells.

The nematic order and the nematic angle of the cells in a cluster shown in *Figure 5F* are calculated as the magnitude and angle of the nematic director defined by

$$\mathbf{n} = \langle (\cos 2\theta_i, \sin 2\theta_i) \rangle_i , \qquad (10)$$

where $\theta_i$ is the angle of the vector $\frac{\mathbf{r}_{i,2} - \mathbf{r}_{i,1}}{|\mathbf{r}_{i,2} - \mathbf{r}_{i,1}|}$ of cell $i$. The average $\langle \cdot \rangle_i$ is calculated over the cells that belong to the cluster.

## Correlation between the cluster elongation and nematic order

The correlation between the cluster elongation and the nematic order of the cells in the cluster (*Figure 5F*, right) is quantified by the order parameter defined by

$$\langle \cos 2\Delta\theta \rangle ,$$

where $\Delta\theta = \theta_l - \theta_n$ is the difference between the longitudinal angle $\theta_l$ and the nematic angle $\theta_n$ of each cluster. Here, note that both angles are of twofold rotational symmetry. To eliminate the effect of small or less-elongated clusters, the average $\langle \cdot \rangle$ is calculated over the clusters composed of more than four cells and the aspect ratio larger than or equal to 2.

## Acknowledgements

We thank Guojun Sheng for critical reading of our manuscript and the members of the Laboratory for Physical Biology for the discussions. This work was supported by Kakenhi grant 16K07385 (YN),

19H14673 (MT) and 22H05170 (TS), JST CREST Grant JPMJCR1852 (TS), and the core funding at RIKEN Center for Biosystems Dynamics Research (TS).

## Additional information

### Funding

| Funder | Grant reference number | Author |
|---|---|---|
| Japan Society for the Promotion of Science | 16K07385 | Yukiko Nakaya |
| Japan Society for the Promotion of Science | 19H14673 | Mitsusuke Tarama |
| Japan Society for the Promotion of Science | 22H05170 | Tatsuo Shibata |
| Japan Science and Technology Agency | 10.52926/JPMJCR1852 | Tatsuo Shibata |
| Japan Society for the Promotion of Science | 19K03645 | Sohei Tasaki |
| Japan Society for the Promotion of Science | 23K03208 | Sohei Tasaki |

The funders had no role in study design, data collection and interpretation, or the decision to submit the work for publication.

### Author contributions

Yukiko Nakaya, Conceptualization, Data curation, Formal analysis, Funding acquisition, Validation, Investigation, Visualization, Methodology, Writing – original draft; Mitsusuke Tarama, Data curation, Software, Formal analysis, Funding acquisition, Validation, Investigation, Visualization, Methodology, Writing – original draft; Sohei Tasaki, Ayako Isomura-Matoba, Investigation; Tatsuo Shibata, Conceptualization, Resources, Data curation, Software, Formal analysis, Supervision, Funding acquisition, Validation, Investigation, Visualization, Methodology, Writing – original draft, Project administration, Writing – review and editing

### Author ORCIDs

Yukiko Nakaya (iD) http://orcid.org/0000-0003-4294-2306
Mitsusuke Tarama (iD) https://orcid.org/0000-0002-2708-1774
Sohei Tasaki (iD) http://orcid.org/0000-0002-4259-0075
Ayako Isomura-Matoba (iD) https://orcid.org/0000-0002-8518-9329
Tatsuo Shibata (iD) https://orcid.org/0000-0002-9294-9998

### Decision letter and Author response

Decision letter https://doi.org/10.7554/eLife.84749.sa1
Author response https://doi.org/10.7554/eLife.84749.sa2

## Additional files

### Supplementary files

MDAR checklist

### Data availability

All source data obtained experimentally during this study are included in the manuscript and supporting files; source data files have been provided for Figures 1-4 and 6. Custom Matlab scripts for the trajectory analysis shown in Figures 1D-J, 4F-K are available at https://doi.org/10.5281/zenodo.17412954. Figure 5 was created using the model detailed in Materials and Methods. All parameter values are shown in Materials and Methods.

The following dataset was generated:

| Author(s) | Year | Dataset title | Dataset URL | Database and Identifier |
|---|---|---|---|---|
| Nakaya Y, Tarama M, Tasaki S, Isomura-Matoba A, Shibata T | 2025 | Migrating mesoderm cells self-organize into a dynamic meshwork structure during chick gastrulation | https://doi.org/10.5281/zenodo.17412954 | Zenodo, 10.5281/zenodo.17412954 |

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
