## [Editor Report]

This important study describes a new mode of collective cell migration during chick embryo development. Quantitative live imaging revealed compelling evidence that cells self-organized into a 3D dynamic meshwork structure while migrating from the epiblast to the endoderm during gastrulation and that this network is associated with N-cadherin-mediated cell-cell adhesion. Agent-based simulations propose that cell-cell adhesions are required for the formation of the meshwork structure and that the cell aspect ratio and cell density may also play a role in the meshwork formation. This manuscript would be of interest to developmental and cell biologists as well as theoreticians studying tissue patterning and collective cell migration.

---

## [Decision Letter]

**Decision letter after peer review:**

Thank you for submitting your article "Migrating mesoderm cells self-organize into a dynamic meshwork structure during chick gastrulation" for consideration by *eLife*. Your article has been reviewed by 3 peer reviewers, and the evaluation has been overseen by a Reviewing Editor and Marianne Bronner as the Senior Editor. The following individual involved in the review of your submission has agreed to reveal their identity: Nicoletta Petridou (Reviewer #3).

Essential Revisions:

1) The authors argue that their agent-based model is used to investigate how the meshwork arises and how it contributes to cell movement. This is not supported by the data. The authors can tone down their claims throughout the manuscript, or provide further support as suggested by the reviewers. Specifically (not all suggestions must be followed, but claims must be supported by data):

a. Analyze how the network structure affects the collective migration of cells (Reviewer #2, point #1).

b. Verify the model prediction regarding the increased rate of mesoderm cells at the primitive streak with a current experimental framework (Reviewer #1).

c. Is the cell's aspect ratio an essential factor for network formation? Reviewer #2 proposes to perform simulations with higher attraction values. (Reviewer #2, point #6)

d. Milder perturbations (e.g., less concentration of N-cad-M so the cells do not lose adhesion completely) or the combination with complementary approaches (e.g., use of blebbistatin, ROCK inhibitor, etc) to understand the contribution of shape, cell-cell adhesion, and density to meshwork formation (Reviewer #3 – point #3)

2) Several clarifications and further quantitative support are required regarding the interpretation of the experimental data. Again, the authors can choose to remove unsupported claims or clarify and provide better support.

a. The argument that mutated cells are excluded from the meshwork is not supported by Figure 4C-D and should be properly quantified (Reviewer #1, Reviewer #2 – point #2).

b. Figure 4J should include the data of all five embryos and the statistical test should take them into consideration (Reviewer #2 – point #4). It is not clear what statistical tests are used throughout the manuscript (e.g., 3/5 embryos usually lead to a p-value > 0.08).

c. There is insufficient data to draw conclusions on the dynamics of the mutant-control cell-cell interactions, this can be resolved with more extensive analysis (Reviewer #2 – point #5).

d. The authors claim that mutated cells are more rounded. This should be quantified (Reviewer #1, Reviewer #2 – point #3)

3) Depleting the intracellular domain of N-cadherin can have effects on intracellular signaling that can lead to changes in cell migration and morphology (cell rounding, de-attachment) which is very much expected to affect the network structure. This is quite a harsh experiment to conclude how N-cadherin contributes to the meshwork, since many parameters change at the same time, and thus these experiments alone do offer much understanding of how the meshwork is formed and functions. In general, the manuscript would benefit a lot from more targeted experiments on either adhesion/shape (different N-cadherin mutants or combinations with other manipulations), and specifically, Reviewer #3 proposes a control experiment of overexpressing N-cad lacking the intracellular domain (Reviewer #3 – point #2). Another possibility is tuning down relevant claims and addressing these issues in the Discussion.

4) Experimental confirmation that the cells being labeled by electroporation (Figure 1) are indeed (mostly) mesodermal and that the meshwork structure is only present in the mesoderm (Figure 2). Reviewer #3 proposes immunostaining or HCR with a mesodermal marker (Reviewer #3 – point #1).

5) The description of the topological data analysis should be elaborated and self-contained. This point was raised by all reviewers. Specifically:

a. Persistent homology analysis: the technical terminology that is inconsistent with the context and very hard to follow. A minimal explanation of the terminology, namely of what is meant by birth-, death rates, and lifetimes, how/why they relate to particular structural properties, and how one should understand the persistence diagrams must be included briefly in the Results and extensively in the Methods section. As a recommendation, Reviewer #2 suggests translating the terminology to be consistent with the context of spatial data (instead of using quantities with temporal terminology).

b. Illustration and a better description of agent-based simulations. The authors should explain how they solved their equations and why they chose a particular potential energy and their set of parameters (Reviewer #2).

6) Regarding the theoretical model, it is not clear to me if the size ratio of hole vs. particles/agents is comparable to the ratio measured experimentally. It seems that in the microscopy images, there are clearly fewer cells surrounding one hole than agents around a hole in the model. A brief quantitative description/analysis would be beneficial to clarify this (Reviewer #3 – point #4).

*Reviewer #1 (Recommendations for the authors):*

Nakaya et al. investigate the cellular mechanisms underlying mesoderm cell migration in chick embryos. Through the use of live and fixed high-resolution imaging and so-called topological data analysis (TDA), they uncover the existence of a dynamic, cell-cell adhesion-dependent meshwork pattern in the mesoderm. By tracking wildtype vs. dominant negative N-cadherin expressing cells, they uncover a dependency of directed collective cell migration but not individual cell migration on cell-cell adhesions. Combining experimental observations with theoretical modeling, the authors demonstrate that elongated shape, attractive interaction, and specific cell densities can sufficiently explain the formation of dynamic meshworks during mesoderm formation.

The key findings in this work are compelling and the integration of theoretical modeling to ask questions that are not experimentally tractable is well-done, done, and well-integrated with experiments. While the author's claims and conclusions are largely justified by their data, a few key observations in the imaging data are described only anecdotally and could be better characterized. Clarifications of some of the methods and motivations would also strengthen the work.

The authors apply topological data analysis (TDA), which is an innovative approach to quantifying tissue topology and provides clear quantitative measures for holes within the meshwork. I appreciate the value of this approach and its application both to experimental and modeling data. However, the authors did not explain this method adequately, and visualization of the results could be improved for clarity.

The use of theoretical modeling and its juxtaposition to experimental data to explain the formation of the meshwork through adhesions, cell shape, and density alone is very strong and compelling, and the data align well with the theoretical model. Since there are numerical predictions that come out of the theoretical models, there are some potential opportunities for experimental measurements that are missed. In fact, while certain elements are quantified very well, some key elements of the model are demonstrated only anecdotal and glossed over – cell shape (shown in supplemental Figure 5 but should be shown earlier), exclusion of dom. Neg. cadherin cells from holes, and time of contact. It would have strengthened the work to better characterize these key observations.

The experiments and modeling of the change in the meshwork structure over time in Figure 6 are compelling. The model predicts an increased rate of appearance of mesoderm cells at the primitive streak. The authors have the opportunity to make an experimental observation of this using their current experimental framework.

Overall, the methods described in this work are likely to have a broad impact in the field of developmental biology in terms of quantitative analysis of imaging data, and a well-designed combination of modeling and experimental results. The biological insights of this work in terms of how the developing mesoderm migrates as a meshwork will likely also open the door for the discovery of such modes of collective migration in other developmental systems.

The use of the classic chick model is not adequately presented. The authors should elaborate in the introduction on the advantages and limitations of the chick model, and how this model has been used historically for such cell migration assays, and how it compares to other experimental models.

Figure 1 does not strongly contribute to the narrative- the anterior/middle/posterior aspect of the mesoderm is not mentioned again, and sometimes taken into consideration and sometimes ignored, even within figure 1. In this reviewer's view, All information in Figure 1 is already contained in Figure 4 in direct contrast to the dom. Neg. cadherin cells, illustrating the desired points much more clearly. Figure 1 panels are also out of order and not referenced in order in the main text, making reading confusing.

For all electroporation experiments- how do the authors know that they are observing mostly mesodermal cells? Are there markers the authors could use to demonstrate this?

Line 135: "The directionality of trajectories was smaller than unity"- what does that mean?

For PD diagrams- for clarity, could the authors create a threshold that indicates a hole and pseudocolor data points in that color?

The authors' claim that the length scale of collective migration is similar to the diameter of the hole is not substantiated.

Clearer control examples of N and p-cadherin endogenous stainings would be useful.

Figure 3: measurements i 3D are not very informative, but if the authors can measure the directionality of the meshwork and how it changes over time (all the inverse xy area for example), or come up with other measures that could indicate dynamics of the network, that would add value.

Last sentence of page 15: "Since the length scales of the polar order of the collective migration and the size of the holes of the meshwork are comparable, we speculate that the frequent rearrangement of the cell-cell contact is a reason why the collective order of the mesoderm cells decays in the long length scale." This reviewer did not follow the logic. If collective order was similar in cells that are near each other if they are constantly rearranging why would that explain this phenomenon?

*Reviewer #2 (Recommendations for the authors):*

The authors provide a detailed experimental and computational investigation of the collective migration of chick mesoderm cells during early gastrulation. Using live imaging and quantitative analysis they characterize the motility of individual cells as they migrate from the primitive streak. From measurements of individual cell directionality, MSD, and cell-speed autocorrelation function they conclude that individual cells migrate in a directed manner but their motion fluctuates in three dimensions and generally appears as a biased random walk. Quantifications of a motility order parameter and measurements of MSRD revealed that the motility of neighboring cells is correlated within a radius of a few tens of microns. Consistent with these findings they also discover that during cell migration a 3D meshwork is formed in between the epiblast and the endoderm. Using persistent homology analysis they characterize the meshwork structure and its dynamics. They find that the meshwork holes are tens of microns in size, consistent with the length scale found in the migration analysis. To investigate the role of cell-cell adhesion in this process they overexpressed an N-cadherin mutant in the mesoderm cells and analyzed cell motility and network formation. Mutated cells were less persistent, the tissue progression speed was lower and their directional coordination was weaker. They, therefore, conclude that cell-cell adhesion is important for the coordinated movement of the cells. To investigate how the meshwork is formed they developed agent-based stochastic simulations of rod-shaped cells interacting via short-range attraction and core repulsion. The simulations showed that meshwork structures form with strong inter-cell attraction and that a threshold cell aspect ratio is required to form the meshwork. In addition, for the networks to form, cell density needed to be sufficiently low since otherwise, they would fill the space with no holes, as they find in later stages of development. The work presented in this manuscript is comprehensive and interesting but I nevertheless have a few important comments.

Strengths: The work provides a detailed characterization of the individual and collective motility of mesoderm cells during gastrulation. The authors discover a novel structure that is formed in the process and present computer simulations that reproduce the structures formed and suggest key factors that may drive and influence their formation.

Weaknesses: Major one: The authors present their analysis of the network structure using persistent homology analysis and use technical terminology that is inconsistent with the context; they describe the structural properties of the meshwork in terms of birth-, death rates, and lifetimes. This makes it very hard to understand their findings. The authors should translate the expressions to the current context, namely use expressions that have concrete structural rather than temporal meaning, and explain in more detail how the points in the persistent diagrams relate to the structural properties of the network.

1. The authors should consider examining their network structure using more direct metrics as done in the analysis of fluid phase transitions in the references mentioned above.

2. The authors could use the simulations described in the section "Dynamic meshwork formation with the supply of agents" to also measure the motility characteristics of the cells in these simulations. This might shed light on how the network structure affects the collective motility of the cells.

Other comments:

1. Abstract line 30-31: The authors say: "To investigate how this meshwork arise and how it contributes to the cell movement we utilized an agent-based theoretical model …". While the authors could use their simulations to explicitly analyze how the network structure affects the collective migration of cells, in the current version of the manuscript this is not done.

2. Line 308 and caption of Figure 4: It isn't clear why the authors say that mutated cells are excluded from the meshwork. They do appear to line a hole. It isn't clear that the images in 4C and 4D shed light on the propensity of the mutated cells to form a network.

3. Line 309-311, 356-358, and 388: Based on 4C and 4D, the authors reach the conclusion that mutated cells are more rounded. This is hard to tell and needs to be quantified.

4. Line 330-334: The graph in 4J should include the data on the two embryos that did not show the effect on Phi(t) and only then the p-value needs to be calculated. It currently is not compelling that there is an effect on the polar order parameter.

5. Line 342-352: The few shown images in Figure 4 supplement 2 are insufficient to draw conclusions on the dynamics of the cell-cell interactions between the mutated and control cells; this requires more extensive analysis.

6. The simulations results appear to resemble a one-component spinodal decomposition of unstable fluids where condensation occurs globally throughout the system (for relevance in biological systems see reviews by Cats and Tailleur, Annu. Rev. Condens. Matter Phys. 2015. 6:219-44 and Joel Berry et al. 2018 Rep. Prog. Phys. 81 046601) It is driven by the attractive van der Waals interactions between particles and does not require them to be anisotropic; for a 2D simulation with Lennard- Jones potential see for instance, Koch et al., Phys Rev A, 1983. I.e., simpler simulations of round particles in attractive interaction (Lennard Jones potential) reveal similar structures in a so-called spinodal decomposition process. Moreover, in the experiments, the cells appear to be quite round. The aspect ratio is close to 1. It is therefore not clear that the rather small cells' aspect ratio is indeed an essential factor for network formation. The authors show in Figure 5C that for a specific choice of cell density and interaction strength, the aspect ratio is important for network formation. I suggest testing stronger attraction strengths (would it not phase separate?) and lower densities.

7. The description of the topological structure analysis in the methods section needs to be self-contained and not be based on the readers' knowledge of the original reference by Obayashi 2018. In addition, as mentioned above, the terminology should also be modified and adapted to the current context of the manuscript and more detailed explanations should be given on how to understand the persistence diagrams (which are a major tool used in the manuscript).

8. Theoretical model section: It would be useful if the authors included an illustration of how their agents and their interaction potential look like. They should add a few sentences explaining how Eq. 1 (and/or 8) was solved, and a few sentences motivating the choice of parameters and of the interaction potential.

*Reviewer #3 (Recommendations for the authors):*

In their manuscript, Nakaya et al. investigate the migration of mesoderm cells in the gastrulating chick embryo to understand how they coordinate as a collective despite the lack of tight confinement and lasting cell contacts. By means of confocal imaging and thorough quantitative analyses for cell tracking, they show that, while individual mesoderm cell motion displays some randomness, the cells move collectively toward the anterior-lateral or lateral direction. Moreover, the mesoderm forms a dynamic 3D meshwork structure, which the authors characterise applying persistent homology analysis. Immunostainings and a knockdown assay hint towards a role of N-cadherin-based cell adhesion in controlling collective cell migration and meshwork structure. Furthermore, Nakaya et al. develop an agent-based theoretical model to recapitulate the experimental observations and test additional parameters, such as cell-cell adhesion, cell elongation, and cell density. This model confirms the importance of adhesion for meshwork formation and supports cell shape and density as further parameters influencing it.

One major strength of this work is the detailed analysis of cellular motion, providing a compelling characterisation of mesoderm cell migration. Moreover, the theoretical model notably complements the experimental observations, while also adding to them. It provides further parameters influencing meshwork formation that could be tested experimentally in the future. Generally, the analysis methods employed in this manuscript constitute a good basis for future work on similar structures and processes. A point of improvement for this study is to understand the differential contribution of N-cadherin in cell adhesion, elongation, and density and through which of these parameters it influences meshwork formation.

All in all, the authors describe a novel mode of collective cell migration of the mesenchymal cells forming the chick mesoderm. This work also contributes to our understanding of cell migration in environments that are not physically restricted and highlights the importance of keeping tissue structure in mind when investigating cellular behaviour. In addition, the methods employed for analysis may be of interest to an audience beyond the field of developmental biology.

Here, I provide some suggestions for experiments that could contribute to strengthening the author's claims, as well as some considerations that in my opinion should be taken into account when coming to conclusions about the experiments:

1. Experiments to confirm that the cells being labelled by electroporation (Figure 1) are indeed (mostly) mesodermal, especially since the authors also suggest other cells may have been labelled, but do not quantify this ("there might have been a few endodermal and epiblast cells", p. 7). This could be done e.g. by immunostaining or HCR with a mesodermal marker. Similarly, in Figure 2, labelling of the epiblast/mesoderm/endoderm via immunostaining or HCR would show in a convincing way that the meshwork structure is only present in the mesoderm.

2. In Figure 4, the authors conclude that intercellular adhesion controls collective mesoderm cell migration based on the effects of overexpression of N-cad-M (lacking the extracellular domain). However, this experimental setup does not take into account that overexpressing the intracellular domain of N-cadherin can have effects on intracellular signalling. Changes in signalling could also account for changes in cell migration, instead of or in addition to the effect of cell adhesion and the associated outcomes of cell elongation. A control experiment would be to overexpress N-cad that lacks the intracellular domain instead and compare the effects between the two manipulations.

3. The theoretical model predicts that shape, cell-cell adhesion, and density contribute to meshwork formation. The authors test these predictions via the N-cad loss of function experiments, where cells naturally round up since they lose cell-cell adhesion to their neighbours. This experimental approach makes it hard to evaluate which of the model parameters are indeed physiologically relevant in vivo. I understand that finding experimental strategies that influence more than one parameter over the other is difficult, however more mild approaches (e.g. less concentration of N-cad-M so the cells do not lose adhesion completely) or the combination with complementary approaches (e.g. use of blebbistatin, ROCK inhibitor, etc) may help understand more the cellular basis of the meshwork formation.

4. Regarding the theoretical model, it is not clear to me if the size ratio of hole vs. particles/agents is comparable to the ratio measured experimentally. It seems that in the microscopy images, there are clearly fewer cells surrounding one hole than agents around a hole in the model. A brief quantitative description/analysis would be beneficial to clarify this.

5. An additional point regarding the theoretical model is if the emergence of the nematic order is also observed in the experimental data, and how this is disrupted in N-cad loss of function.

6. Overall, the manuscript is clearly written, with only some minor typos ("chamotaxis" in Table S6, "transvers" in Figure 2 S1A). Additionally, the sentence "… five embryos, each of which contains a lot of cells" (p. 18) could be rephrased to be more specific regarding the approximate number of cells.

7. The figures are clear and well-structured, with some sketches greatly contributing to the understanding of the experimental setups and imaging regions. However, in some cases, displaying the fluorescent images in different colours could make them clearer (e.g. green and cyan together do not provide much contrast; green and red in the same image is not accessible to people with red-green colour blindness, as in Figure 3, Figure 3 S1, Figure 4, Figure 4 S2). Additionally, in Figure 1C it may be helpful to clarify that the dot marks the initial position of the cells (as is done in Figure 4E). In Figure 4, Figure 4 S1 and S2, as well as in the main text, more consistency in the naming of the mutant N-cadherin construct would add cohesion to the work ("N-Cad M", "Ncad DN", "N-cad mutant", "N-cad-M", "N-Cad-M").

[Editors' note: further revisions were suggested prior to acceptance, as described below.]

Thank you for resubmitting your work entitled "Migrating mesoderm cells self-organize into a dynamic meshwork structure during chick gastrulation" for further consideration by *eLife*. Your revised article has been evaluated by Marianne Bronner (Senior Editor) and a Reviewing Editor.

The manuscript has been improved but there are some remaining issues that need to be addressed, namely toning down claims that are not fully supported by the data (or providing the data to support these claims). Please refer carefully and fully to the comments by reviewers 2 and 3.

*Reviewer #1 (Recommendations for the authors):*

The authors have addressed all concerns to this reviewer's satisfaction. This reviewer has no further suggestions.

*Reviewer #2 (Recommendations for the authors):*

The authors adequately responded to all my major comments. Nevertheless, with one point I think the authors need to tone done their conclusion. Although with their theoretical model the authors did not observe network formation with round cells, this does not yet mean that cell elongation is an essential factor for network formation in the experimental system. Indeed, as mentioned in my previous report, the network structures found in their experiments resemble the patterns formed with unstable fluids undergoing a one-component spinodal decomposition. This process is driven by the attractive van der Waals interactions between particles and does not require them to be anisotropic in shape. See references in my previous report. This implies that network formation of the type that the authors find could indeed form with interacting round cells under appropriate conditions.

*Reviewer #3 (Recommendations for the authors):*

The authors have not addressed most of the essential revisions asked.

Even if some of the experiments did not work, it is recommended that the authors provide those data for the reviewers to be able to access the results. If this is not possible, the abstract should substantially change, because the necessary data to fully support the conclusions are missing.

Some examples:

(1) Confirm that the cells being labelled by electroporation are mesodermal cells. The new figures make it easier to understand but there is no quantification and if the authors conclude that is not possible to know if these are mesodermal cells, then they cannot fully support their conclusion that this is a mechanism of mesoderm-specific cell migration.

(2) N-cadherin manipulations / rhok inhibitors. Since no other alternative experiments were possible by the authors, the conclusions should change. Especially in the abstract and not only briefly in the discussion

---

## [Author Response]

Essential Revisions:1) The authors argue that their agent-based model is used to investigate how the meshwork arises and how it contributes to cell movement. This is not supported by the data. The authors can tone down their claims throughout the manuscript, or provide further support as suggested by the reviewers. Specifically (not all suggestions must be followed, but claims must be supported by data):a. Analyze how the network structure affects the collective migration of cells (Reviewer #2, point #1).

The mean squared displacement was of course measured for the simulation results, which showed super diffusive property, like the experimental results. But this did not change significantly for varied values of the attraction and aspect ratio. Thus, unfortunately no clear conclusion on the motility was drawn.

b. Verify the model prediction regarding the increased rate of mesoderm cells at the primitive streak with a current experimental framework (Reviewer #1).

Not all mesoderm cells can be labeled by the electroporation. Additionally, the rate of cell labeling varies from experiment to experiment. It is also difficult to perform a long timelapse observation including multiple stages. For these reasons, measuring of increased rate of mesoderm cells at the primitive streak remains a challenge for future research. In response to the comments, we have adjusted the expression to weaken the claim. The revised statement now is: "From these results, one possible reason of the decrease of the hole size as the developmental stage of the embryo proceeds could be a potential increase in the appearance rate of mesoderm cells at the primitive streak.”

c. Is the cell's aspect ratio an essential factor for network formation? Reviewer #2 proposes to perform simulations with higher attraction values. (Reviewer #2, point #6)

As shown in Figures 5A-B, the agent-based simulation exhibits percolation at higher densities, where phase separation occurs between the denser region and the voids. However, there is a critical aspect ratio below which the size of voids becomes very small even in the percolation regime. The percolation is promoted when the attractive interaction is introduced, but for a very high attraction the critical density for percolation becomes large again (See Author response image 1). Therefore, the cell’s aspect ratio is an essential factor for the network formation. For stronger attraction strength and lower density, percolation does not occur and the network structure was not observed.

**Author response image 1. sa2fig1:** Percolation transition happens as the density increases. The critical density becomes smaller when the attractive interaction is introduced, but it becomes large again for too high attraction.

d. Milder perturbations (e.g., less concentration of N-cad-M so the cells do not lose adhesion completely) or the combination with complementary approaches (e.g., use of blebbistatin, ROCK inhibitor, etc) to understand the contribution of shape, cell-cell adhesion, and density to meshwork formation (Reviewer #3 – point #3)

We recognize the importance of conducting experiments to quantitatively assess the effects of disrupting cell-cell adhesion on the meshwork formation, such as low concentration of N-Cad-M. However, we would like to draw your attention to some of the practical challenges we face in carrying out the suggested experiment. Despite our best efforts, the experimental success rate of introducing plasmids into a sufficient number of cells has been low, making it challenging to generate a robust dataset for quantitative evaluation. Given the inherent complexities, conducting a substantial number of experiments to achieve statistical significance appears to be impractical within the constraints of our current resources. This difficulty was one reason why we developed a theoretical model and performed simulations. Therefore, we concluded that the quantitative experimental approaches to better understand the cellular basis of the meshwork formation remain to be a future issue.

2) Several clarifications and further quantitative support are required regarding the interpretation of the experimental data. Again, the authors can choose to remove unsupported claims or clarify and provide better support.a. The argument that mutated cells are excluded from the meshwork is not supported by Figure 4C-D and should be properly quantified (Reviewer #1, Reviewer #2 – point #2).

Visual inspection suggested a tendency for cells expressing N-Cad-M to be excluded from the meshwork. However, it was challenging to quantitatively define the holes that constitute the meshwork. Without clear definition of these holes, quantitatively assessing the exclusion of cells proved difficult. To address this, we employed persistent homology, which allows for quantitative characterization of the meshwork. Although the use of persistent homology was considered appropriate, the low proportion of cells expressing N-Cad-M limited the robustness of quantification. Therefore, we decided to soften the statement as indicated in the revised manuscript as “Indeed, the cells expressing N-CadM tended to be somewhat excluded from the meshwork of the control cells (Figure 4D).”

b. Figure 4J should include the data of all five embryos and the statistical test should take them into consideration (Reviewer #2 – point #4). It is not clear what statistical tests are used throughout the manuscript (e.g., 3/5 embryos usually lead to a p-value > 0.08).

In the revised manuscript, we clearly indicate that the difference in the polar order parameter between control and mutated cells is NOT statistically significant. The statistical tests we used in this manuscript is indicated in the figure legends. For Figure 4J (Figure 4F-K), paired t-test was used to compare the average values obtained from five embryos. The obtained p-values were indicated in individual graphs (Figure 4F-K). The p-value for Figure 4J was 0.081, which indicated that the difference was not statistically significant.

In Figure 4 figure supplement 1 D, the temporal average of the polar order parameter in the areas of 125μm x 125μm were indicated for control and mutated cells. The average order parameters were different in three embryos among five embryos according to Wilcoxon rank sum test between the control and mutant cells.

c. There is insufficient data to draw conclusions on the dynamics of the mutant-control cell-cell interactions, this can be resolved with more extensive analysis (Reviewer #2 – point #5).

We appreciate the reviewer’s concern that the data shown in Figure 4—figure supplement 2 may be insufficient to conclusively characterize the dynamics of cell-cell interactions between control and mutant cells. We acknowledge that a more extensive quantitative analysis would be necessary to rigorously establish differences in intercellular contact behavior.

Given our limitations in image acquisition, we have modified the main text to clarify that these observations are qualitative and speculative rather than conclusive. Specifically, we now state that the time-lapse images suggest that mutant cells exhibit reduced contact duration, which may contribute to their more randomized movement. However, we note that a more comprehensive dataset would be required to statistically validate this trend.

We appreciate this feedback, as it has helped us refine the presentation of our findings and ensure that our claims are appropriately framed.

Here is the revised version mentioning these results in the main text:

“To explore potential differences in intercellular contact dynamics between control and mutant cells, we carefully examined time-lapse images. Qualitative observations suggest that control cells tend to elongate their bodies and establish prolonged cell-cell contacts via protrusions (Figure 4—figure supplement 3). These cells remained in contact for several tens of minutes, with the longest contact duration exceeding one hour (No.1 and No.2 pair in Figure 4—figure supplement 3, upper panels, Video 6). In contrast, mutant cells appeared to exhibit shorter-lived interactions, as they typically did not maintain cellcell contacts for more than 20 minutes after colliding with neighboring cells (See No.1 cell in Figure 4—figure supplement 3, bottom panels, Video 6). While these observations suggest that reduced contact duration may contribute to more randomized movement of mutant cells, leading to frequent changes in the direction of collective migration, a more extensive quantitative analysis would be necessary to confirm this trend.”

d. The authors claim that mutated cells are more rounded. This should be quantified (Reviewer #1, Reviewer #2 – point #3)

We quantified the cell aspect ratio in Figure 4 figure supplement 1. We mentioned this point in the main text (line 429-431). According to the quantification, “the aspect ratio of mesoderm cells was 2.34 ± 0.08 (± SEM) (Figure 4 figure supplement 1, control), while that of the N-cadherin mutant cells was 1.91 ± 0.08 (± SEM) (Figure 4 figure supplement 1, N-Cad-M).”

Before this quantification, we also mentioned in the main text that the mutated cells were more rounded (line 338-340, 384-386, 416-418). We have toned down the assertion that the mutated cells are more rounded in shape and have referenced the figure for quantification (Figure 4 figure supplement 1).

3) Depleting the intracellular domain of N-cadherin can have effects on intracellular signaling that can lead to changes in cell migration and morphology (cell rounding, de-attachment) which is very much expected to affect the network structure. This is quite a harsh experiment to conclude how N-cadherin contributes to the meshwork, since many parameters change at the same time, and thus these experiments alone do offer much understanding of how the meshwork is formed and functions. In general, the manuscript would benefit a lot from more targeted experiments on either adhesion/shape (different N-cadherin mutants or combinations with other manipulations), and specifically, Reviewer #3 proposes a control experiment of overexpressing N-cad lacking the intracellular domain (Reviewer #3 – point #2). Another possibility is tuning down relevant claims and addressing these issues in the Discussion.

Since the deletion mutant of N-cadherin (N-Cad-M) lacks the extracellular domain that is responsible for cell-cell adhesion activity, it is expected that expression of N-Cad-M primarily decreases cell-cell adhesion. However, as pointed out by the reviewer, within the current data we cannot exclude the possibility that the introduction of N-Cad-M may affect other processes such as cell motility. Therefore, to mitigate the claim that “cell-cell adhesion” affects the collective cell migration, we modify the statement to suggest that N-Cad-M has an effect on the collective cell migration. In addition, potential effects of N-Cad-M on the collective cell migration other than cell-cell adhesion are discussed in the Discussion (line 552-557).

4) Experimental confirmation that the cells being labeled by electroporation (Figure 1) are indeed (mostly) mesodermal and that the meshwork structure is only present in the mesoderm (Figure 2). Reviewer #3 proposes immunostaining or HCR with a mesodermal marker (Reviewer #3 – point #1).

We looked at the expression of a typical mesoderm marker Brachury. The expression levels were quite variable (see Author response image 2 showing the Brachury expression level at different z level). Even in the mesoderm region (B, horizontal section), some cells expressed it and others did not (D). It is also expressed in some cells in the epiblast (A horizontal and C transverse section). This may be due to the fact that the embryos we studied were at a relatively early stage (HH4). Therefore, we concluded that it is not possible to test whether cells expressing H2B-eGFP introduced by electroporation are mesoderm by co-expression of Brachury.

**Author response image 2. sa2fig2:** Brachury expression level at different z level of early chick embryo.

Next, we fixed an embryo with H2B-eGFP introduced by electroporation after the timelapse imaging and stained it with DAPI and Phalloidin to determine the position where the cells expressing H2B-eGFP were located (Figure 1 figure supplement 1A). The transverse section of the embryos showed that almost all the cells expressing H2B-eGFP were in the region of mesoderm between the epiblast and the endoderm. Therefore, from this histological examination, almost all the cells were determined to be mesoderm.To avoid confusion, we removed the sentence mentioning the possibility of including epiblast and endoderm, and made the following correction: “Almost all of these 3D trajectories were of mesoderm cells (Figure 1 figure supplement 1A).”

5) The description of the topological data analysis should be elaborated and self-contained. This point was raised by all reviewers. Specifically:a. Persistent homology analysis: the technical terminology that is inconsistent with the context and very hard to follow. A minimal explanation of the terminology, namely of what is meant by birth-, death rates, and lifetimes, how/why they relate to particular structural properties, and how one should understand the persistence diagrams must be included briefly in the Results and extensively in the Methods section. As a recommendation, Reviewer #2 suggests translating the terminology to be consistent with the context of spatial data (instead of using quantities with temporal terminology).

We apologize for any confusion caused by not providing a clear explanation of the concept of persistent homology. In the revised manuscript, the method of persistent homology was concisely explained using the analogy of an image as a landscape. The terminology of persistent homology, such as birth and death times and lifetime, also caused confusion. We will replace the terms "birth time”, “death time” and “lifetime” with “birth level”, “death level” and “persistence,” respectively.

Explanations of persistent homology added to the main text are as follows: “To analyze an image by persistent homology, we first prepare a black and white binary pixel image from the original image by thresholding, where the pixels occupied by cells are white. Such a binary image can be seen as a landscape with hills and valleys, where hills represent cluster of cells, and the valleys represent holes between them. We can assign height and depth (negative height) of hills and valleys, respectively, by considering the shortest distance (Manhattan distance) from the interface between white and black pixels. Now, we consider this landscape at different height. To uncover topological features, we apply progressive thresholding to the binary image from the minimum height (leading to completely white) to the maximum (leading to completely black) in a stepwise manner. At each threshold level, pixels below the threshold are black. As the threshold level incrementally changes, a feature, such as a hole which is a black area surrounded by a white area, emerges or “is born”. Such a threshold level is recorded as “birth level” of the feature. As the threshold level is increased further, the feature merges with another feature or “dies”. Such a threshold level is recorded as “death level” of the feature. Thus, the holes are characterized by the two quantities called birth and death levels, and they are visualized by points in persistence diagram (PD) where the coordinates are given by the birth and death levels (Figure 2F). These two quantities basically represent the size of the feature and the distance between two features, respectively. The difference between the death and the birth levels is called “persistence”, which becomes large for a reliable topological feature. Note that in the persistent homology on binary images, the birth and death levels, and thus the persistence, are measured in the unit of pixels.”

Persistent homology can filter out noise with small persistence while capturing topological features with large persistence, the method is robust with respect to noise. Thus, the persistent homology is a reliable tool for analyzing biological images with inherent noise. We were exploring methods to quantify meshwork structure from cell-tocell connections. However, due to the noisy nature of the images, automatically achieving this from the images was challenging. Meshwork structure is considered to be a structure that can only be identified by some kind of appropriate coarse-graining. Persistent homology provide a suitable approach for this purpose. We hope that the revise manuscript gives a better explanation of how persistent homology works to identify the characteristic structure that an image contains, and how it is robust against the noise.

b. Illustration and a better description of agent-based simulations. The authors should explain how they solved their equations and why they chose a particular potential energy and their set of parameters (Reviewer #2).

The equations of motion (Eq.1) were solve numerically using the Euler method, which is now indicated in the method section. The value of the attractive interaction is chosen to be an optimal value where percolation is most promoted (See Author response image 1). The cell width is used as the unit of length; stretching elasticity is set large enough so that the shape is kept almost constant; interaction length is set to half the cell width so that the cells interact only with the nearest neighbours; effective viscosity is set as that of water; temperature is set to room temperature. We also provide the schematics of the model and the potential and force profiles of the attractive interaction in Figure 5 figure supplement 1.

6) Regarding the theoretical model, it is not clear to me if the size ratio of hole vs. particles/agents is comparable to the ratio measured experimentally. It seems that in the microscopy images, there are clearly fewer cells surrounding one hole than agents around a hole in the model. A brief quantitative description/analysis would be beneficial to clarify this (Reviewer #3 – point #4).

We understand the reviewer’s concern regarding the difference in hole size between the simulation and the experiment and appreciate the opportunity to clarify this point. In response, we would like to emphasize that we have already quantified the number of cells surrounding the holes in Figure 6C and 6D (right y-axis). Our analysis indicates that hole size is indeed smaller in the experiment than in the simulation, as stated in the original main text:

“The radius of holes obtained from the PDs (Figure 6B bottom) decreases as well (Figure 6D), although the hole sizes in the simulation were slightly larger than those of the experiment. This quantitative difference might come from the fact that the shape of the cells in the simulation is kept constant, while the shape of the real mesoderm cells changes dynamically.”

In the revised manuscript, we will explicitly state the number of surrounding cells as follows:

“The radius of holes obtained from the PDs (Figure 6B bottom) decreases as well (Figure 6D), although the hole sizes and the number of cells surrounding the holes (Figure 6C and D, left y-axis) in the simulation were slightly larger than those of the experiment.”

This revision ensures that the quantitative comparison is explicitly mentioned, addressing the reviewer’s concern.

Reviewer #1 (Recommendations for the authors):Nakaya et al. investigate the cellular mechanisms underlying mesoderm cell migration in chick embryos. Through the use of live and fixed high-resolution imaging and so-called topological data analysis (TDA), they uncover the existence of a dynamic, cell-cell adhesion-dependent meshwork pattern in the mesoderm. By tracking wildtype vs. dominant negative N-cadherin expressing cells, they uncover a dependency of directed collective cell migration but not individual cell migration on cell-cell adhesions. Combining experimental observations with theoretical modeling, the authors demonstrate that elongated shape, attractive interaction, and specific cell densities can sufficiently explain the formation of dynamic meshworks during mesoderm formation.

We thank the reviewer for the clear summary and positive assessment of our work. We are pleased that the reviewer appreciates our combined use of high-resolution imaging, topological data analysis, and theoretical modeling to reveal a novel, cell-cell adhesion– dependent mode of collective mesoderm migration in chick embryos. We are encouraged by the recognition of our findings regarding the role of N-cadherin and the conditions required for meshwork formation.

The key findings in this work are compelling and the integration of theoretical modeling to ask questions that are not experimentally tractable is well-done, done, and well-integrated with experiments. While the author's claims and conclusions are largely justified by their data, a few key observations in the imaging data are described only anecdotally and could be better characterized. Clarifications of some of the methods and motivations would also strengthen the work.

We thank the reviewer for their positive comments on the integration of theoretical modeling with experimental data, and for recognizing the overall strength of our conclusions. In response to the reviewer’s suggestion, we have revised the text to clarify several methodological details and motivations behind our analyses, such as topological data analysis (TDA) and use of chick embryos. While we agree that certain imaging observations—such as cell shape, and contact duration—are important, we note that limitations in image quality or resolution made robust quantification difficult in some cases. For such instances, we have softened the corresponding statements in the manuscript to avoid overinterpretation and to maintain appropriate rigor.

The authors apply topological data analysis (TDA), which is an innovative approach to quantifying tissue topology and provides clear quantitative measures for holes within the meshwork. I appreciate the value of this approach and its application both to experimental and modeling data. However, the authors did not explain this method adequately, and visualization of the results could be improved for clarity.

We thank the reviewer for recognizing the value of applying topological data analysis (TDA) in our study. In the revised manuscript, we have substantially improved the explanation of TDA in the main text to ensure better clarity and accessibility. Specifically, we refined several terminologies to reduce potential confusion and make the description more intuitive for readers unfamiliar with the method. See 5a in Essential Revisions.

The use of theoretical modeling and its juxtaposition to experimental data to explain the formation of the meshwork through adhesions, cell shape, and density alone is very strong and compelling, and the data align well with the theoretical model. Since there are numerical predictions that come out of the theoretical models, there are some potential opportunities for experimental measurements that are missed. In fact, while certain elements are quantified very well, some key elements of the model are demonstrated only anecdotal and glossed over – cell shape (shown in supplemental Figure 5 but should be shown earlier), exclusion of dom. Neg. cadherin cells from holes, and time of contact. It would have strengthened the work to better characterize these key observations.

We thank the reviewer for their thoughtful comments and for highlighting the strength of the theoretical model and its alignment with the experimental data. We agree that certain aspects of the model—such as cell shape, exclusion of dominant-negative N-cadherin– expressing cells from the meshwork, and the duration of cell-cell contact—could further benefit from more direct experimental characterization. However, as mentioned above, limitations in imaging resolution and sample size made it challenging to robustly quantify these features. For this reason, we chose to present them cautiously and have softened the corresponding claims in the revised manuscript. In addition, we have moved the relevant image showing cell aspect ratio (previously in Figure 5—figure supplement 1) to an earlier position (Figure 4—figure supplement 1) according to the suggestion. We hope that these revisions provide a clearer and more balanced presentation of the data.

The experiments and modeling of the change in the meshwork structure over time in Figure 6 are compelling. The model predicts an increased rate of appearance of mesoderm cells at the primitive streak. The authors have the opportunity to make an experimental observation of this using their current experimental framework.

We thank the reviewer for their positive remarks on the time-resolved analysis of meshwork structure and the predictive value of the model shown in Figure 6. We agree that the predicted increase in the appearance rate of mesoderm cells at the primitive streak is an interesting and testable feature. While our current imaging framework captures the overall dynamics of the meshwork, it was not optimized for quantitatively tracking cell ingress rates at the primitive streak over time. We consider this an important future direction.

Overall, the methods described in this work are likely to have a broad impact in the field of developmental biology in terms of quantitative analysis of imaging data, and a well-designed combination of modeling and experimental results. The biological insights of this work in terms of how the developing mesoderm migrates as a meshwork will likely also open the door for the discovery of such modes of collective migration in other developmental systems.

We sincerely thank the reviewer for their encouraging comments regarding the broader impact of our methodological approach and the biological insights provided by this study. We are especially pleased that the potential relevance of meshwork-based collective migration to other developmental systems was recognized. We hope that our work will contribute to further exploration of collective behaviors in embryonic tissues and inspire new directions in both experimental and theoretical developmental biology.

The use of the classic chick model is not adequately presented. The authors should elaborate in the introduction on the advantages and limitations of the chick model, and how this model has been used historically for such cell migration assays, and how it compares to other experimental models.

We would like to thank the reviewer for the helpful comment to improve our manuscript. Now, we have included the works on mesoderm cell migration in different model systems and advantage of the use of chick model in Introduction as follows.

“Mesoderm migration have been studied mostly in fly, fish, frog, chick and mouse embryos. However, spatiotemporal patterns of migration differ among the models. In zebrafish prechordal plate, cells exhibit collective movement towards the animal pole, whose directionality is maintained by contact between cells (Dumortier et al., 2012). In contrast, in mice, live imaging of whole embryo indicated that the mesoderm cells migrate individually rather than collective (Ichikawa et al., 2013). More recently, in mice, embryonic mesoderm cells demonstrate random aspects in their migration behavior and neighboring cells tend to migrate in parallel (Saykali et al., 2019). Leveraging the flat and easily accessible nature of the early chick embryo, researchers have conducted extensive and well-documented experimental studies over many years. The chick embryo has been particularly valuable in studying the properties of the mesoderm, including the migration pathways that depend on cell fate (Psychoyos and Stern, 1996). Live imaging techniques have been also employed to visualize the migratory routes of cells emerging from various positions along the primitive streak, spanning from anterior to posterior regions (Yang et al., 2002).”

Figure 1 does not strongly contribute to the narrative- the anterior/middle/posterior aspect of the mesoderm is not mentioned again, and sometimes taken into consideration and sometimes ignored, even within figure 1. In this reviewer's view, All information in Figure 1 is already contained in Figure 4 in direct contrast to the dom. Neg. cadherin cells, illustrating the desired points much more clearly. Figure 1 panels are also out of order and not referenced in order in the main text, making reading confusing.

Figure 1 is one of the most important results of this paper. Cell tracking was performed in much wider area than that in Figure 4. Using this data, basic cell motility analysis has been performed, and its properties have been shown. It also features simultaneous cell tracking in anterior, middle, posterior regions. We found that the properties of cell migration did not differ significantly between these three regions. We have carefully checked the main text to ensure that the panels in Figure 1 are referenced in the correct order. As reviewer #3 said that “the figures are clear and well-structured” in the comment #7, we think that Figure 1 gives fundamental information about the mesoderm cell migration in the early chick embryo.

For all electroporation experiments- how do the authors know that they are observing mostly mesodermal cells? Are there markers the authors could use to demonstrate this?

Please see our response to 4 in Essential Revisions.

Line 135: "The directionality of trajectories was smaller than unity"- what does that mean?

We have rephrased this sentence as follows. “The directionality of trajectories was smaller than one”.

For PD diagrams- for clarity, could the authors create a threshold that indicates a hole and pseudocolor data points in that color?

We have indicated the threshold values to define holes in the Method section of the original manuscript. The holes are identified as the points in the persistence diagram with the birth level smaller than -10 µm and the death level larger than -2.5 µm. Only for Figure 6C, we choose the points with the birth level smaller than -5 µm and the death level larger than -2.5 µm because there was no hole satisfying the above stricter threshold for the later stage. We have indicated these threshold values in the figure legends. The color in the persistence diagram indicates the multiplicity of the points.

The authors' claim that the length scale of collective migration is similar to the diameter of the hole is not substantiated.

According to our cell tracking analysis, the characteristic length of the collective migration is about 60 µm, indicating that the cells within the range of 60 µm tend to migrate in the similar direction. The average diameter of the holes was estimated to be 34 µm. Because there is no order of magnitude difference between the characteristic length of the collective migration and the diameter of the holes, we considered that the two sizes are almost comparable to each other. To describe the difference more precisely, we change the expression as follows:

“Thus, the diameter of the hole is about half of the characteristic length scale of collective migration measured in the previous subsection (~60 µm).”

Clearer control examples of N and p-cadherin endogenous stainings would be useful.

We have shown the images of endogenous N- and P-cadherin in Figures 4A and C.

Figure 3: measurements i 3D are not very informative, but if the authors can measure the directionality of the meshwork and how it changes over time (all the inverse xy area for example), or come up with other measures that could indicate dynamics of the network, that would add value.

In Figure 3D, we wanted to show the appearance, disappearance, fusion and fission of holes. We considered plotting the trajectories of center of holes, but could not come up with a good way to represent fusion and fission processes. Quantifying directionality was not done because the number of samples from which hole dynamics were measured was small that it would not provide statistically meaningful data.

Last sentence of page 15: "Since the length scales of the polar order of the collective migration and the size of the holes of the meshwork are comparable, we speculate that the frequent rearrangement of the cell-cell contact is a reason why the collective order of the mesoderm cells decays in the long length scale." This reviewer did not follow the logic. If collective order was similar in cells that are near each other if they are constantly rearranging why would that explain this phenomenon?

In the subsection that contains the sentence pointed out by the reviewer, we found that the meshwork is quite dynamic in the sense that cells forming a hole are replaced over time due to frequent changes in the relative positions among cells. We also found in the previous subsections that the length scales of the collective order and the diameter of the holes are comparable. The sentence was attempting to explain what can be speculated from these findings. Since the previous expression was insufficient, we have replaced it with the following sentences:

Since the characteristic length scale of collective migration and the diameter of the holes are comparable, the cells that form a hole move in roughly the same direction. However, frequent changes in the relative positions among the cells cause the collective order to decay over long length scales much larger than the hole size.

Reviewer #2 (Recommendations for the authors):The authors provide a detailed experimental and computational investigation of the collective migration of chick mesoderm cells during early gastrulation. Using live imaging and quantitative analysis they characterize the motility of individual cells as they migrate from the primitive streak. From measurements of individual cell directionality, MSD, and cell-speed autocorrelation function they conclude that individual cells migrate in a directed manner but their motion fluctuates in three dimensions and generally appears as a biased random walk. Quantifications of a motility order parameter and measurements of MSRD revealed that the motility of neighboring cells is correlated within a radius of a few tens of microns. Consistent with these findings they also discover that during cell migration a 3D meshwork is formed in between the epiblast and the endoderm. Using persistent homology analysis they characterize the meshwork structure and its dynamics. They find that the meshwork holes are tens of microns in size, consistent with the length scale found in the migration analysis. To investigate the role of cell-cell adhesion in this process they overexpressed an N-cadherin mutant in the mesoderm cells and analyzed cell motility and network formation. Mutated cells were less persistent, the tissue progression speed was lower and their directional coordination was weaker. They, therefore, conclude that cell-cell adhesion is important for the coordinated movement of the cells. To investigate how the meshwork is formed they developed agent-based stochastic simulations of rod-shaped cells interacting via short-range attraction and core repulsion. The simulations showed that meshwork structures form with strong inter-cell attraction and that a threshold cell aspect ratio is required to form the meshwork. In addition, for the networks to form, cell density needed to be sufficiently low since otherwise, they would fill the space with no holes, as they find in later stages of development. The work presented in this manuscript is comprehensive and interesting but I nevertheless have a few important comments.Strengths: The work provides a detailed characterization of the individual and collective motility of mesoderm cells during gastrulation. The authors discover a novel structure that is formed in the process and present computer simulations that reproduce the structures formed and suggest key factors that may drive and influence their formation.Weaknesses: Major one: The authors present their analysis of the network structure using persistent homology analysis and use technical terminology that is inconsistent with the context; they describe the structural properties of the meshwork in terms of birth-, death rates, and lifetimes. This makes it very hard to understand their findings. The authors should translate the expressions to the current context, namely use expressions that have concrete structural rather than temporal meaning, and explain in more detail how the points in the persistent diagrams relate to the structural properties of the network.

We thank the reviewer for their comprehensive and thoughtful evaluation of our work, and for acknowledging both the novelty of our findings and the integration of experimental and computational approaches. We are encouraged by the reviewer’s recognition of the detailed characterization of mesodermal cell motility, the discovery of the 3D meshwork structure, and the modeling that recapitulates the key aspects of the system.

We particularly appreciate the reviewer’s comment regarding the use of persistent homology analysis. In the original version, we indeed used standard terminology from topological data analysis (such as “birth time” “death time” and “lifetime”) without sufficiently translating these terms into the biological context of our study. In the revised manuscript, we have addressed this issue by replacing those terms with “birth level”, “death level” and “persistence”, respectively. (See our response to 5a in Essential Revisions). We have substantially updated the explanation about topological data analysis in the main text. We have also revised the figure legends (Figure 2 and 3) to clarify how points in the persistence diagrams correspond to holes.

1. The authors should consider examining their network structure using more direct metrics as done in the analysis of fluid phase transitions in the references mentioned above.

Our aim was to quantitatively characterize the holes that appear in the inhomogeneous distribution of cells. Initially, we explored several conventional methods for this purpose. Specifically, we attempted to extract cell nucleus positions and infer cellular connectivity based on nucleus-to-nucleus distances. However, due to the anisotropic elongation of cell shapes, nucleus distances did not reliably reflect cell proximity, making it difficult to characterize hole formation. Similarly, approaches based on Delaunay triangulation were also challenging, as they assume isotropic spacing between points, which is not valid in our case.

We also considered image-based thresholding techniques to directly extract hole regions. However, setting a single threshold value to robustly detect hole locations proved to be highly challenging due to variations in cell density and morphology across different regions.

Given these challenges, we adopted topological data analysis (TDA), which allows us to quantify topological features such as holes without relying on predefined connectivity assumptions or arbitrary thresholds. Unlike conventional network-based approaches, TDA provides a multi-scale characterization of spatial structures, making it particularly well-suited for analyzing the distribution of non-uniformly arranged cells with irregular shapes.

We acknowledge the importance of conventional network metrics. However, a systematic comparison between different methods for characterizing network structures is beyond the scope of this paper.

2. The authors could use the simulations described in the section "Dynamic meshwork formation with the supply of agents" to also measure the motility characteristics of the cells in these simulations. This might shed light on how the network structure affects the collective motility of the cells.

The mean squared displacement was of course measured for the simulation results, which showed super diffusive property, like the experimental results. But this did not change significantly for varied values of the attraction and aspect ratio. Thus, unfortunately no clear conclusion on the motility was drawn.

Other comments:1. Abstract line 30-31: The authors say: "To investigate how this meshwork arise and how it contributes to the cell movement we utilized an agent-based theoretical model …". While the authors could use their simulations to explicitly analyze how the network structure affects the collective migration of cells, in the current version of the manuscript this is not done.

We appreciate the reviewer’s comment regarding the role of the theoretical model in analyzing collective cell migration. The primary aim of our modeling approach was to understand how cell elongation, cell-cell adhesion strength, and cell density contribute to the emergence of a dynamic meshwork. The simulation results successfully reproduced hole formation, collapse, and rearrangement, consistent with our experimental observations.

While our findings strongly suggest that the formation and maintenance of the meshwork structure play a crucial role in facilitating collective migration by enabling transient but coordinated interactions between cells, we acknowledge the reviewer’s concern. In response, we have revised the abstract to better reflect the primary focus of the model, emphasizing the key factors underlying meshwork formation and dynamic reorganization, and have removed the phrase “how it contributes to the cell movement” from the abstract.

2. Line 308 and caption of Figure 4: It isn't clear why the authors say that mutated cells are excluded from the meshwork. They do appear to line a hole. It isn't clear that the images in 4C and 4D shed light on the propensity of the mutated cells to form a network.

Our statement that the mutant cells are excluded from the meshwork does not imply that they are completely physically isolated within the tissue but rather that they fail to integrate into the cell-cell adhesion network that defines the meshwork structure.

To clarify this point, we will revise the relevant text to explicitly state that “N-Cad-Mexpressing cells tend not to integrate into the meshwork structure of control mesoderm cells” instead of “excluded from the meshwork.” We appreciate this opportunity to refine our explanation.

3. Line 309-311, 356-358, and 388: Based on 4C and 4D, the authors reach the conclusion that mutated cells are more rounded. This is hard to tell and needs to be quantified.

We appreciate the reviewer’s suggestion regarding the need for quantification to support our conclusion that mutant cells are more rounded. Figure 4—figure supplement 1 provides a quantitative comparison of the aspect ratio between control and N-Cad-Mexpressing cells, demonstrating that mutant cells have a lower aspect ratio, confirming their more rounded morphology. To improve clarity, we will remove the reference to Figure 4C and 4D in this context, as these images alone do not sufficiently convey this difference without quantification.

4. Line 330-334: The graph in 4J should include the data on the two embryos that did not show the effect on Phi(t) and only then the p-value needs to be calculated. It currently is not compelling that there is an effect on the polar order parameter.

We appreciate the reviewer’s concern regarding the discussion of the polar order parameter in Figure 4J. While the graph already included data from all five embryos with the p-value (0.081), we acknowledge that the original draft primarily discussed only three of them, which may have led to an incomplete representation of the overall trend.

In response, we have updated the text to incorporate the results from all five embryos to ensure consistency with the data shown in the figure. Additionally, given that two embryos did not show a clear effect, we have weakened our statement in the main text to reflect this variability. Specifically, we now state that “While the polar order parameter \begin{document}$\varphi\left(t \right)$\end{document} on the mutant side tended to be smaller than that on the control side, the overall difference was not statistically significant (Figure 4J and Figure 4 figure supplement 1D).”

This revision ensures that our discussion accurately represents the full dataset and prevents any unintended overinterpretation of the results.

5. Line 342-352: The few shown images in Figure 4 supplement 2 are insufficient to draw conclusions on the dynamics of the cell-cell interactions between the mutated and control cells; this requires more extensive analysis.

Please see our response to 2-c in Essential Revisions.

6. The simulations results appear to resemble a one-component spinodal decomposition of unstable fluids where condensation occurs globally throughout the system (for relevance in biological systems see reviews by Cats and Tailleur, Annu. Rev. Condens. Matter Phys. 2015. 6:219-44 and Joel Berry et al. 2018 Rep. Prog. Phys. 81 046601) It is driven by the attractive van der Waals interactions between particles and does not require them to be anisotropic; for a 2D simulation with Lennard- Jones potential see for instance, Koch et al., Phys Rev A, 1983. I.e., simpler simulations of round particles in attractive interaction (Lennard Jones potential) reveal similar structures in a so-called spinodal decomposition process. Moreover, in the experiments, the cells appear to be quite round. The aspect ratio is close to 1. It is therefore not clear that the rather small cells' aspect ratio is indeed an essential factor for network formation. The authors show in Figure 5C that for a specific choice of cell density and interaction strength, the aspect ratio is important for network formation. I suggest testing stronger attraction strengths (would it not phase separate?) and lower densities.

Please see our response to 1-c in Essential Revisions.

7. The description of the topological structure analysis in the methods section needs to be self-contained and not be based on the readers' knowledge of the original reference by Obayashi 2018. In addition, as mentioned above, the terminology should also be modified and adapted to the current context of the manuscript and more detailed explanations should be given on how to understand the persistence diagrams (which are a major tool used in the manuscript).

Please see our response to (5)-a in Essential Revisions.

8. Theoretical model section: It would be useful if the authors included an illustration of how their agents and their interaction potential look like. They should add a few sentences explaining how Eq. 1 (and/or 8) was solved, and a few sentences motivating the choice of parameters and of the interaction potential.

Please see our response to (5)-b in Essential Revisions.

Reviewer #3 (Recommendations for the authors):In their manuscript, Nakaya et al. investigate the migration of mesoderm cells in the gastrulating chick embryo to understand how they coordinate as a collective despite the lack of tight confinement and lasting cell contacts. By means of confocal imaging and thorough quantitative analyses for cell tracking, they show that, while individual mesoderm cell motion displays some randomness, the cells move collectively toward the anterior-lateral or lateral direction. Moreover, the mesoderm forms a dynamic 3D meshwork structure, which the authors characterise applying persistent homology analysis. Immunostainings and a knockdown assay hint towards a role of N-cadherin-based cell adhesion in controlling collective cell migration and meshwork structure. Furthermore, Nakaya et al. develop an agent-based theoretical model to recapitulate the experimental observations and test additional parameters, such as cell-cell adhesion, cell elongation, and cell density. This model confirms the importance of adhesion for meshwork formation and supports cell shape and density as further parameters influencing it.One major strength of this work is the detailed analysis of cellular motion, providing a compelling characterisation of mesoderm cell migration. Moreover, the theoretical model notably complements the experimental observations, while also adding to them. It provides further parameters influencing meshwork formation that could be tested experimentally in the future. Generally, the analysis methods employed in this manuscript constitute a good basis for future work on similar structures and processes. A point of improvement for this study is to understand the differential contribution of N-cadherin in cell adhesion, elongation, and density and through which of these parameters it influences meshwork formation.All in all, the authors describe a novel mode of collective cell migration of the mesenchymal cells forming the chick mesoderm. This work also contributes to our understanding of cell migration in environments that are not physically restricted and highlights the importance of keeping tissue structure in mind when investigating cellular behaviour. In addition, the methods employed for analysis may be of interest to an audience beyond the field of developmental biology.

We thank the reviewer for their thoughtful and encouraging evaluation of our manuscript. We are grateful for the recognition of our detailed quantitative analysis of mesodermal cell motion and the identification of a novel mode of collective migration through a dynamic 3D meshwork structure. We also appreciate the reviewer’s positive comments on the complementarity of the theoretical model and the broader relevance of our methodological approach.

Regarding the reviewer’s suggestion to further dissect the role of N-cadherin in terms of its potential influence on cell adhesion, elongation, and density, we fully agree that this is an important and insightful point. While our current data clearly show that N-Cad-M expression impairs directional coordination and meshwork integrity, experimentally separating its specific contributions to adhesion strength, cell morphology, and effective cell density within the tissue remains technically challenging. We have added a comment in Discussion (lines 553-557) to acknowledge this limitation and to propose it as an important direction for future studies, possibly involving tools that allow more selective modulation of these parameters.

Here, I provide some suggestions for experiments that could contribute to strengthening the author's claims, as well as some considerations that in my opinion should be taken into account when coming to conclusions about the experiments:1. Experiments to confirm that the cells being labelled by electroporation (Figure 1) are indeed (mostly) mesodermal, especially since the authors also suggest other cells may have been labelled, but do not quantify this ("there might have been a few endodermal and epiblast cells", p. 7). This could be done e.g. by immunostaining or HCR with a mesodermal marker. Similarly, in Figure 2, labelling of the epiblast/mesoderm/endoderm via immunostaining or HCR would show in a convincing way that the meshwork structure is only present in the mesoderm.

Please see our response to 4 in Essential Revisions.

2. In Figure 4, the authors conclude that intercellular adhesion controls collective mesoderm cell migration based on the effects of overexpression of N-cad-M (lacking the extracellular domain). However, this experimental setup does not take into account that overexpressing the intracellular domain of N-cadherin can have effects on intracellular signalling. Changes in signalling could also account for changes in cell migration, instead of or in addition to the effect of cell adhesion and the associated outcomes of cell elongation. A control experiment would be to overexpress N-cad that lacks the intracellular domain instead and compare the effects between the two manipulations.

Please see our response to 3 in Essential Revisions.

3. The theoretical model predicts that shape, cell-cell adhesion, and density contribute to meshwork formation. The authors test these predictions via the N-cad loss of function experiments, where cells naturally round up since they lose cell-cell adhesion to their neighbours. This experimental approach makes it hard to evaluate which of the model parameters are indeed physiologically relevant in vivo. I understand that finding experimental strategies that influence more than one parameter over the other is difficult, however more mild approaches (e.g. less concentration of N-cad-M so the cells do not lose adhesion completely) or the combination with complementary approaches (e.g. use of blebbistatin, ROCK inhibitor, etc) may help understand more the cellular basis of the meshwork formation.

Please see our response to 1d in Essential Revisions.

4. Regarding the theoretical model, it is not clear to me if the size ratio of hole vs. particles/agents is comparable to the ratio measured experimentally. It seems that in the microscopy images, there are clearly fewer cells surrounding one hole than agents around a hole in the model. A brief quantitative description/analysis would be beneficial to clarify this.

Please see our response to 6 in Essential Revisions.

5. An additional point regarding the theoretical model is if the emergence of the nematic order is also observed in the experimental data, and how this is disrupted in N-cad loss of function.

We appreciate the reviewer’s suggestion to assess whether the emergence of nematic order in our theoretical model is also observed in the experimental data and how this is affected by N-cadherin loss of function. In response, we conducted an additional analysis of nematic order in our experimental dataset.

Ideally, nematic order should be defined in terms of cell shape anisotropy. However, extracting cell shape anisotropy from enough samples to statistically assess nematic order was challenging. Therefore, we instead evaluated nematic order based on the polarity of cell motion. While we observed a statistically significant decrease in nematic order in 2 out of 5 embryos, the difference was not statistically significant when considering all samples.

One possible explanation for this variability is that N-cadherin disruption was not fully efficient in all embryos. Variations in the extent of N-cadherin loss could influence the degree to which nematic order is affected, potentially contributing to the lack of a consistent trend across samples. Additionally, since motion-based polarity may not fully capture the underlying structural order of the cell arrangement, this methodological limitation may have further contributed to the inconsistency in our analysis.

Given these considerations, the variability in N-cadherin disruption efficiency and the challenge of accurately assessing nematic order from available data, we have decided not to include this analysis in the revised manuscript, as the current data do not provide a clear or consistent conclusion. However, we appreciate this valuable suggestion and acknowledge that future studies with a larger sample size, improved N-cadherin disruption, and more direct measurements of shape anisotropy-based nematic order may be needed to further investigate this relationship.

6. Overall, the manuscript is clearly written, with only some minor typos ("chamotaxis" in Table S6, "transvers" in Figure 2 S1A). Additionally, the sentence "… five embryos, each of which contains a lot of cells" (p. 18) could be rephrased to be more specific regarding the approximate number of cells.

We have corrected the points raised.

7. The figures are clear and well-structured, with some sketches greatly contributing to the understanding of the experimental setups and imaging regions. However, in some cases, displaying the fluorescent images in different colours could make them clearer (e.g. green and cyan together do not provide much contrast; green and red in the same image is not accessible to people with red-green colour blindness, as in Figure 3, Figure 3 S1, Figure 4, Figure 4 S2). Additionally, in Figure 1C it may be helpful to clarify that the dot marks the initial position of the cells (as is done in Figure 4E). In Figure 4, Figure 4 S1 and S2, as well as in the main text, more consistency in the naming of the mutant N-cadherin construct would add cohesion to the work ("N-Cad M", "Ncad DN", "N-cad mutant", "N-cad-M", "N-Cad-M").

We thank this review for pointing out that the figures are clear and well-structured. According to the suggestion, we have changed the red color to magenta in Figures 3, 3S1, 4 and 4S2. For Figure 1C, we mention in the legend that the dot indicates the initial position of the trajectories, as “The initial position of the cells is marked by dots on the trajectories.” We have consistently used N-Cad-M as an abbreviation for mutant N-cadherin, according to the suggestion.

[Editors’ note: what follows is the authors’ response to the second round of review.]

The manuscript has been improved but there are some remaining issues that need to be addressed, namely toning down claims that are not fully supported by the data (or providing the data to support these claims). Please refer carefully and fully to the comments by reviewers 2 and 3.Reviewer #2 (Recommendations for the authors):The authors adequately responded to all my major comments. Nevertheless, with one point I think the authors need to tone done their conclusion. Although with their theoretical model the authors did not observe network formation with round cells, this does not yet mean that cell elongation is an essential factor for network formation in the experimental system. Indeed, as mentioned in my previous report, the network structures found in their experiments resemble the patterns formed with unstable fluids undergoing a one-component spinodal decomposition. This process is driven by the attractive van der Waals interactions between particles and does not require them to be anisotropic in shape. See references in my previous report. This implies that network formation of the type that the authors find could indeed form with interacting round cells under appropriate conditions.

Visual inspection alone cannot determine whether the cell-cell adhesion mediated by N-cadherin can actually induce a spinodal-like instability. In principle, one could test for such an instability by quantifying domain coarsening, but this is impractical here because cells are continuously supplied from the primitive streak, and the supply rate increases over time, masking any intrinsic coarsening law. As we cannot entirely rule out the possibility, we now acknowledge it at the end of Discussion. If such an instability exists, anisotropic cell shape would act on top of the instability process. The inserted paragraph follows.

“Our data and theoretical model suggest that cell elongation and cell-cell adhesion actively stabilize the meshwork. There may still be a possibility that a spinodal-like instability driven by N-cadherin-mediated adhesion. If it exists, anisotropic cell shape would act on top of the instability process. Future work will be required to determine whether such an instability exists in the meshwork formation of chick mesoderm.”

From our analysis of cell motility, the Péclet number is estimated to be 2 to 4, given that the mean cell speed is from 2 to 4 µm/min, typical cell size is about 10 µm, and the persistent time obtained from the MSRD analysis is about 10 min. This value is much smaller than the value required for the motility induced phase separation (MIPS). Thus, we consider MIPS unlikely to drive meshwork formation.

Reviewer #3 (Recommendations for the authors):The authors have not addressed most of the essential revisions asked.Even if some of the experiments did not work, it is recommended that the authors provide those data for the reviewers to be able to access the results. If this is not possible, the abstract should substantially change, because the necessary data to fully support the conclusions are missing.Some examples:(1) Confirm that the cells being labelled by electroporation are mesodermal cells. The new figures make it easier to understand but there is no quantification and if the authors conclude that is not possible to know if these are mesodermal cells, then they cannot fully support their conclusion that this is a mechanism of mesoderm-specific cell migration.(2) N-cadherin manipulations / rhok inhibitors. Since no other alternative experiments were possible by the authors, the conclusions should change. Especially in the abstract and not only briefly in the discussion

We agree with the reviewer that some of the interpretation should be presented more cautiously given the limitations of our validation experiments. In response, we have revised the abstract and main text. For (1), in the abstract, we refrained from using the term “mesoderm cells”. Instead, we described the labeled population as cells emerging from the primitive streak and migrating between ectoderm and endoderm. In the main texts, we explicitly note that a small fraction of definitive endoderm cells may be included. For (2), in the abstract, since from our experiment it might not be straight forward to conclude how N-cadherin contributes to the meshwork, we removed the sentence asserting that the meshwork structure was supported by Ncadherin. Throughout the manuscript we now frame N-cadherin as a potential contributor to the collective cell migration. The revised abstract follows.

“Migration of cell populations is a fundamental process in morphogenesis and disease. The mechanisms of collective cell migration of epithelial cell populations have been well studied. It remains unclear, however, how the highly motile mesenchymal cells, which migrate extensively throughout the embryo, are connected with each other and coordinated as a collective. During chick gastrulation, cells emerging from the primitive streak and migrating in the 3D space between ectoderm and endoderm (mesoderm region) exhibit a novel form of collective migration. Using live imaging and quantitative analysis, such as topological data analysis (TDA), we found that these cells undergo a novel form of collective migration, in which they form a meshwork structure while moving away from the primitive streak. Overexpressing a mutant form of N-cadherin was associated with reduced speed of tissue progression and directionality of the collective cell movement, whereas the speed of individual cells remains unchanged. To investigate how this meshwork arises, we utilized an agent-based theoretical model, suggesting that cell elongation, cell-cell adhesion, and cell density are the key parameters for the meshwork formation. These data provide novel insights into how a supracellular structure of migrating mesenchymal cells may arise in loosely connected cell populations.”